



# Challenges in predicting Greenland supraglacial lake drainages at the regional scale

Kristin Poinar[1] and Lauren C. Andrews[2]

[1]University at Buffalo, Department of Geology
[2]NASA Goddard Space Flight Center, Global Modeling and Assimilation Office

**Correspondence:** K. Poinar (kpoinar@buffalo.edu)

**Abstract.** A leading hypothesis for the mechanism of fast supraglacial lake drainages is that transient extensional stresses briefly allow crevassing in otherwise-compressive ice flow regimes. Lake water can then hydrofracture the crevasse to the base of the ice sheet, and river inputs can maintain this connection as a moulin. If future ice-sheet models are to accurately represent moulins, we must understand their formation processes, timescales, and locations. Here, we use remote-sensing

velocity products to constrain the relationship between strain rates and lake drainages across ∼1600 km² in Pâkitsoq, western Greenland, between 2002–2019. We find significantly more-extensional background strain rates at moulins associated with fast-draining lakes than at slow-draining or non-draining lake moulins. We test whether moulins in more-extensional background settings drain their lakes earlier, but find insignificant correlation. To investigate the frequency that strain-rate transients are associated with fast lake drainage, we examined Landsat-derived strain rates over 16- and 32-day periods at moulins associated

with 240 fast lake drainage events over 18 years. A low signal-to-noise ratio, the presence of water, and the multi-week repeat cycle obscured any resolution of the hypothesized transient strain rates. Our results support the hypothesis that transient strain rates drive fast lake drainages. However, the current generation of ice-sheet velocity products, even when stacked across hundreds of fast lake drainages, cannot resolve these transients. Thus, observational progress in understanding lake drainage initiation will rely on field-based tools such as GPS networks and photogrammetry.

## 1 Introduction

Nearly all surface meltwater produced in the ablation zone of the western Greenland Ice Sheet descends to the ice-sheet bed through moulins, vertical englacial conduits that deliver surface melt to the subglacial environment (Smith et al., 2015). The water flux through moulins drives the evolution of the subglacial hydrologic system on daily and seasonal timescales (Schoof, 2010; Hoffman et al., 2011; Hewitt, 2013; Werder et al., 2013; Andrews et al., 2014). This, in turn, is a crucial control on the

20 flow of the ice above (Iken and Bindschadler, 1986; Bartholomew et al., 2012) Our ability to predict future ice-sheet mass balance in western Greenland thus depends, among other things, on understanding the patterns of meltwater delivery to the bed through moulins (Banwell et al., 2016; Gagliardini and Werder, 2018).



## 1.1 Need for prediction of lake-drainage dates

The subglacial hydrological system evolves in response to two primary aspects of surface meltwater input to the ice-sheet bed:
(1) the locations where moulins deliver surface water to the bed, and (2) the temporal evolution, including diurnal to seasonal variability, of that water flux to the bed (Schoof, 2010; Banwell et al., 2016). Moulins are readily identified in high-resolution airborne (Thomsen, 1988) or satellite (Phillips et al., 2011; Andrews, 2015; Smith et al., 2015; Yang and Smith, 2016) imagery, often via supraglacial stream terminations. In addition, many moulins lie within or just outside basins of specific supraglacial lakes. Meltwater collects in these lakes through the melt season and may abruptly drain to the bed, presumably reactivating an existing moulin or forming a new moulin (Das et al., 2008; Catania et al., 2008; Banwell et al., 2012; Stevens et al., 2015; Chudley et al., 2019). The timing of the reactivation or creation of individual lake-draining moulins can thus be determined by using satellite imagery to identify the dates of disappearance of supraglacial lakes (McMillan et al., 2007; Sundal et al., 2009; Selmes et al., 2011; Liang et al., 2012; Morriss et al., 2013; Johansson et al., 2013; Fitzpatrick et al., 2014; Williamson et al., 2017).

Once a lake-draining moulin is activated, it will generally carry surface meltwater to the base of the ice sheet for the duration of the melt season (Chudley et al., 2019). The volume and flow rate of water through the moulin to the ice-sheet bed vary in time; this is a crucial driver of the evolution of the subglacial hydrological system on both daily and seasonal timescales (Schoof, 2010). Thus, the variation in time and space of the surface water supply to the bed is an important input for models of subglacial hydrology, and, in turn, for ice-dynamics models (Banwell et al., 2016; Sommers et al., 2018). Currently, such basal input hydrographs can be constructed through observational techniques such as lake and stream water depth estimation from satellite imagery (Box and Ski, 2007; Sneed and Hamilton, 2007; Georgiou et al., 2009; Morriss et al., 2013; Fitzpatrick et al., 2014; Pope et al., 2016; Moussavi et al., 2016), supraglacial stream gauging and routing routines (Smith et al., 2015, 2017), and catchment-scale analysis of regional climate model output (Banwell et al., 2012; Rennermalm et al., 2013; van As et al., 2014), as long as moulin locations are known.

Current remote-sensing techniques can locate moulins, identify the date that a lake-draining moulin initiates water delivery to the bed, and quantify the water flux that a moulin carries to the bed. Our ability to generalize the patterns of moulin formation in space and time, however, is limited by our developing understanding of the sequence of events that cause lake drainages, particularly at the regional scale. In particular, a quantitative description of moulin location and activation date is currently lacking. Such a parameterization could be used to construct multi-year, regional-scale basal input hydrographs, which the next generation of ice-sheet models will require (Banwell et al., 2016; Sommers et al., 2018). Here, we work to identify controls on the drainage dates of supraglacial lakes so that the locations and timing of seasonal water input to the bed across the ablation zone of western Greenland can be parameterized in future climate scenarios.

## 1.2 Previous work on triggers of supraglacial lake drainage

In western Greenland, lake-draining moulins are formed or re-activated by meltwater amassed in supraglacial lakes (Das et al., 2008; Tedesco et al., 2013; Stevens et al., 2015); other moulins likely form by the interaction of supraglacial streams with





crevasses, absent lakes (McGrath et al., 2011; Smith et al., 2015; Koziol et al., 2017). In either case, formation of a moulin requires formation of a fracture, which generally requires an extensional strain rate regime (Alley et al., 2005). Remote-sensing surveys over western Greenland, however, have found that supraglacial lakes generally occupy areas where the annual mean strain rates are compressional (Joughin et al., 2013; Poinar et al., 2015; Andrews, 2015; Catania et al., 2008). This implies

that supraglacial lake drainage and the associated development of moulins is caused by transient perturbations in the local strain-rate regime (Christoffersen et al., 2018; Hoffman et al., 2018; Stevens et al., 2015).

  Recent studies have greatly advanced our understanding of the mechanism of moulin formation. Specifically, high-magnitude, short-duration perturbations to the background strain rate have been observed in the hours before and during rapid drainages of supraglacial lakes (Stevens et al., 2015; Hoffman et al., 2018; Christoffersen et al., 2018). The origins of these perturbations

are not fully certain, but there is good evidence that temporary local "blisters" of subglacial water (Boon and Sharp, 2003; Tedesco et al., 2013; Stevens et al., 2015; Chudley et al., 2019) or differential ice acceleration due to changing subglacial water inputs in the vicinity of the lake (Christoffersen et al., 2018) can temporarily push the ice under a supraglacial lake into an extensional strain-rate regime, which could incite ice fracture near or within the supraglacial lake, allowing a moulin to form. However, the ubiquity of these strain-rate transients, also called lake drainage precursors, are not known. For instance, Stevens

et al. (2015) observed such precursors in three melt seasons at a single fast-draining lake, but Chudley et al. (2019) did not. We seek to investigate the presence or absence of these precursors at the regional scale.

  In this work, we test two related hypotheses: (1) that background ice-sheet conditions, including annual-average strain rates, affect the timing of lake drainage, and (2) that transient high-magnitude strain rates, or lake-drainage precursors, precede fast supraglacial lake drainage at lakes across the ablation zone. We focus on 78 previously identified lakes in the Pâkitsoq area, a

well-studied and accessible region on the western Greenland Ice Sheet.

## 2 Methods

We examine time-varying principal strain rates at moulin sites on the ice-sheet surface in an effort to quantify their influence on the date that a moulin begins delivering significant meltwater fluxes to the bed. We equate this "moulin activation" with the date of rapid supraglacial lake drainage, which we identify at 78 lake sites in 2010–2018, by analyzing time series of two visible

satellite imagery products, following Selmes et al. (2011). In addition, we use WorldView imagery to identify the locations of the moulins that drained each lake, in years and at locations where imagery is available. We calculate principal strain rates at these locations from multiple velocity products with varying spatial resolution, temporal resolution, and precision. Finally, we compare the lake drainage dates to the local strain rate time series at the moulin sites.

### 2.1 Velocity products

We analyzed the posted east-west ($u$) and north-south ($v$) velocities, as well as their respective errors $\delta u$ and $\delta v$, of the regional velocity products shown in Fig. 1. These include NASA MEaSUREs Multi-year Greenland Ice Sheet Velocity Mosaic, Version 1 (Joughin et al., 2016a, b), the MEaSUREs Greenland Ice Sheet Velocity Map from InSAR Data, Version 2 (Joughin et al.,



2018b, 2010), European Space Agency (ESA) Climate Change Initiative (Boncori et al., 2018), the GoLIVE Landsat-8-derived velocity product (Scambos et al., 2016; Fahnestock et al., 2016), and a similar Landsat-derived product from the Technical

University Dresden (Rosenau et al., 2015).

### 2.1.1   Radar-derived winter velocities

We use the NASA MEaSUREs Multi-year Greenland Ice Sheet Velocity Mosaic, Version 1 (Joughin et al., 2016a, b) to study the long-term average strain rates at moulins. This dataset comprises representative velocities in Pâkitsoq across the twenty-year period 1995–2015 and is derived from multiple InSAR satellite sensors, using speckle tracking and interferometry. The

product is supplemented with Landsat 8 data at limited times (2014–2015 only) to improve accuracy in regions with limited InSAR coverage. The data represent year-round velocities, rather than isolating summer or winter (Joughin et al., 2016a). The pixel size $\Delta x = \Delta y$ is 250 m, and typical uncertainties in our study region are ∼20 m/yr or ∼20% of the average velocity.

We use the MEaSUREs Greenland Ice Sheet Velocity Map from InSAR Data, Version 2 (Joughin et al., 2018b, 2010) to study wintertime strain rates at moulins. This product is derived from wintertime SAR acquisitions to create flow maps for individual

winters. As with the multi-year dataset above, these velocity data originate from speckle tracking and interferometry. Figure 1 shows the temporal extent of our use of this dataset: we use data from all available winters from 2005/2006 through 2017/2018. The pixel size $\Delta x = \Delta y$ is 500 m for winters 2005–2012 and 200 m for winters 2012–2018. Typical uncertainties in our study region are ∼7 m/yr or ∼9% of the average velocity.

As shown in Fig. 1, the MEaSUREs maps contain a handful of gaps: winters 2010/2011, 2011/2012, 2013/2014, and

2017/2018. To fill the 2013/2014 gap, we used a dataset from the European Space Agency Climate Change Initiative (Boncori et al., 2018; Khvorostovsky et al., 2016). Velocities from that winter were derived from intensity-tracking of RADARSAT-2 data (Nagler et al., 2016). The pixel size $\Delta x = \Delta y$ is 500 m and average uncertainties in our study region are ∼35 m/yr, or ∼20% of the average velocity.

### 2.1.2   Landsat-derived melt season velocities

Recent development and application of image processing tools to Landsat Operational Land Imager (OLI) image pairs are now producing ice flow maps for glaciers and ice sheets (Heid and Kääb, 2012). These velocity datasets are spatially continuous and relatively high-resolution, with pixel sizes on the order of hundreds of meters (typically 10×10 to 25×25 Landsat OLI Band 8 pixels, or 150–375 m), and time spacing in 16-day multiples, equivalent to the Landsat repeat cycle (Rosenau et al., 2015; Fahnestock et al., 2016). Data generation requires two cloud-free daytime acquisitions that have good image correlation; thus,

these maps can have incomplete coverage, with data missing from a number of pixels. We use two Landsat-based ice velocity products: the Technical University of Dresden (TUD) Landsat-7-derived product (Rosenau et al., 2015) and the GoLIVE Landsat-8-derived product (Fahnestock et al., 2016).

To study strain rates in melt seasons 2002–2015, we used the Technical University of Dresden (TUD) product (Rosenau et al., 2015). Temporal coverage of the TUD velocity product is 1985–2015 (Fig. 1), and we used velocity observations constructed

from Landsat scenes spaced by ≤32 days. Spatial coverage is limited to elevations below ∼900 m, which includes 18 of the



78 lakes in our study area. The TUD velocity product pixel size $\Delta x = \Delta y$ is 225 m (15×15 Landsat OLI Band 8 pixels) and average uncertainties in our study region are ∼140 m/yr, or ∼70% of the average velocity. For additional quality control, we discarded observations at all pixels whose flow directions deviated more than 90° from the wintertime flow direction, which we defined from the twenty-year average (Joughin et al., 2016a, b), which amounted to 5% of all observations. Overall, we used

a subset of the TUD dataset that contained 181 velocity maps associated with 97 fast-draining lake events at 18 low-elevation lakes over the 14-year period 2002–2015.

To study strain rates in melt seasons 2013–2019, we used the GoLIVE product (Scambos et al., 2016; Fahnestock et al., 2016) (Fig. 1). We selected velocity observations constructed from Landsat scenes spaced by 16 days. This differs from our 16–32 day choice for the TUD velocities for two reasons: (1) There were more GoLIVE observations available overall, allowing

tighter constraints in image spacing; and (2) including 32-day observations significantly smoothed any strain rate signal. This dataset covers our entire study area. The GoLIVE pixel size $\Delta x = \Delta y$ is 300 m (20×20 Landsat OLI Band 8 pixels) and average uncertainties in our study region are ∼55 m/yr, or ∼30% of the average velocity. As with the TUD velocities, we discarded observations with flow directions >90° from background (Joughin et al., 2016a, b), which here amounted to 3% of all observations. Thus, we used 97 velocity maps associated with 152 fast-draining lake events at 59 distinct lakes over the

7-year period 2013–2019.

## 2.2   Calculation of principal strain rates

We analyzed each velocity dataset described above separately. For each dataset, we smoothed the velocities with a 1 km × 1 km boxcar filter, which carries through many high-strain-rate features (King, 2018). We propagated errors in the velocity observations through this filtering using the following formula:

$$\delta_{v-filt} = \frac{1}{n}\sqrt{\Sigma_i^n \left(\delta_{v_i}\right)^2} \tag{1}$$

for an $n$-element filter (in the case of our 3×3 filter, $n = 9$), for both components of the surface velocity $v$ (eastward and northward). We then calculated the two horizontal principal strain rates $\dot{\epsilon}_1$ and $\dot{\epsilon}_3$ by taking centered spatial derivatives of these smoothed velocities, as follows:

$$\dot{\epsilon}_1 = \frac{1}{2}\left(\frac{\partial u}{\partial x} + \frac{\partial v}{\partial y}\right) - \frac{1}{2}\sqrt{\left(\frac{\partial u}{\partial x} - \frac{\partial v}{\partial y}\right)^2 + \left(\frac{\partial u}{\partial y} + \frac{\partial v}{\partial x}\right)^2} \tag{2}$$

$$\dot{\epsilon}_3 = \frac{1}{2}\left(\frac{\partial u}{\partial x} + \frac{\partial v}{\partial y}\right) + \frac{1}{2}\sqrt{\left(\frac{\partial u}{\partial x} - \frac{\partial v}{\partial y}\right)^2 + \left(\frac{\partial u}{\partial y} + \frac{\partial v}{\partial x}\right)^2} \tag{3}$$

These definitions follow Harper and Humphrey (Harper et al., 1998). We used twice the native pixel size, $2\Delta x$ and $2\Delta y$, as the differential spatial length, effectively averaging strain rates across three adjacent pixels in each direction. We assumed plane



strain and thus approximated the third principal strain rate, $\dot{\epsilon}_2$, which points in the vertical direction, as zero. Uncertainties in the strain rates are derived from basic error propagation principles and are given by

$$\delta_{\dot{\epsilon}_{1,3}} = \frac{1}{2\Delta x}\sqrt{\left(\delta_u\right)^2 + \left(\delta_v\right)^2} \tag{4}$$

in the case where $\Delta x = \Delta y$, which is generally true for remote-sensing data. A more general expression for strain rate uncertainties is obtained in the case where $\Delta x \neq \Delta y$, which is applicable to field-derived data such as that obtained from GPS networks:

$$\delta_{\dot{\epsilon}_{1,3}} = \sqrt{\frac{1}{8}\left[\left(\frac{\partial u}{\partial x}\right)^2 + \left(\frac{\partial v}{\partial y}\right)^2\right] + \frac{1}{8}\left(\frac{\left(\frac{\partial u}{\partial x} - \frac{\partial v}{\partial y}\right)^2\left[\left(\frac{\partial u}{\partial x}\right)^2 + \left(\frac{\partial v}{\partial y}\right)^2\right] + \left(\frac{\partial u}{\partial y} - \frac{\partial v}{\partial x}\right)^2\left[\left(\frac{\partial u}{\partial y}\right)^2 + \left(\frac{\partial v}{\partial x}\right)^2\right]}{\left(\frac{\partial u}{\partial x} + \frac{\partial v}{\partial y}\right)^2 + \left(\frac{\partial u}{\partial y} + \frac{\partial v}{\partial x}\right)^2}\right)} \tag{5}$$

## 155   2.3   Identification of lake drainage dates from satellite imagery

We identify the drainage dates of 78 lakes in our study area from remote sensing imagery over the melt seasons 2010–2019, following the methods of Selmes et al. (2011). We supplement our dataset with drainage dates for the same 78 lakes over 2006–2009 from Morriss et al. (2013).

### 2.3.1   Optical imagery datasets

To maximize our confidence in identifying when lakes drained, we analyzed both MODIS and Landsat images, leveraging the higher temporal resolution of MODIS alongside the finer spatial resolution of Landsat. We used the red and infrared bands because they provide the greatest contrast between lake pixels (liquid water) and non-lake pixels (snow or bare ice) (Liang et al., 2012).

We used MODIS images collected by the Aqua satellite, which passes overhead at 1:30 pm local time, which is a good
approximation of the expected lake drainage time at peak melt in the early afternoon (Das et al., 2008). We supplemented Aqua data with data from the Terra satellite (10:30 am pass) for the years 2011, 2017, and 2018, in part due to a greater than average number of cloud-obscured Landsat acquisitions in these years. We accessed the MODIS data through Google Earth Engine image collections (Gorelick et al., 2017), which are pre-assembled datasets of scenes with near-nadir looks. We use surface reflectance bands 1 and 2 (620–670 nm red wavelength and 841–876 nm infrared wavelength) at 250-m spatial
resolution (Vermote and Wolfe, 2015d, b). We supplemented the 250-m product with the red band (620–670 nm) of the 500-m product (Vermote and Wolfe, 2015c, a). We also used the 500-m product to construct a cloud mask to apply to all bands. We rejected pixels identified as "cloud" or "cloud shadow", fields which are posted at 1 km resolution. We also rejected pixels that do not have the highest quality in band 3 (blue), which is posted at 500 m. Thus, we retained all 250-m pixels within larger 1-km pixels identified as highest-quality. This approach differs from other studies that selected completely cloud-free MODIS





scenes (Box and Ski, 2007; Sundal et al., 2009); our approach should yield fewer missing dates and thus higher temporal resolution for each lake.

We also analyzed reflectance from Landsat images. We used the red band (640–670 nm, 30-m spatial resolution) and did not apply any cloud mask. We primarily used scenes from Rows 8–10, Path 11, which are the descending (daytime), passes of our study area. We supplemented these with three scenes from the ascending (nighttime) passes, Rows 81–82, Path 233, to fill 180 cloudy gaps in 2018. The high latitude of our study area means that the ascending scenes are illuminated in the summer.

### 2.3.2 Method of discrimination of lake drainage type and date

We analyzed the 78 Pâkitsoq lake locations identified by Morriss et al. (2013) over eight summers, 2010–2018. For each lake location, we calculated the ratio of reflectance averaged over the 1×1-km box centered on the lake location, to that averaged over the 2×2-km square ring that surrounding the central box, following Selmes et al. (2011). We calculated this ratio for all 185 three band-products (MODIS red, MODIS infrared, and Landsat red) to create a time series for each lake, for June 1 through August 31 of each year. In years with earlier or later melt seasons, we extended the time series through May 1 and September 30, as needed. Figure 2 shows these ratios. Next, we assessed the three reflectance-ratio time series at each lake as well as close-ups of the 5–12 Landsat scenes (Fig. 2). As a lake grows over the course of a melt season, the lake-to-surrounding-ice reflectance ratio we calculated generally decreases, as more-absorptive water gradually fills the central 1×1-km box. When a lake drains, 190 the more-reflective ice-sheet surface is exposed in the central box, returning the reflectance ratio to approximately 1. The newly exposed lake-bottom ice can be more reflective than the surrounding ice (Darnell et al., 2013), making the reflectance ratio of drained lakes sometimes exceed 1. We manually identified the most likely dates over which each lake drained by assessing these curves, and assigned each lake to a drainage category: rapidly draining, slowly draining, refreezing, or uncertain. We discriminated rapidly draining lakes from slowly draining lakes by the length of time it took their reflectance ratios to return 195 to near 1 from their minima (Fig. 2). When this time is ≤ 6 days, we classified a lake as fast-draining; when it is >6 days, we classified the lake as slow-draining. This threshold falls within the range of 2–6 days established in the literature (Selmes et al., 2011; Morriss et al., 2013; Selmes et al., 2013; Fitzpatrick et al., 2014; Williamson et al., 2017; Miles et al., 2017; Koziol et al., 2017; Williamson et al., 2018a). We choose 6 days for agreement with Morriss et al. (2013), whose 2002–2010 data we extend forward through 2019.

We also rated the completeness of drainage of each fast- or slow-draining lake by analyzing the amount of water left in the basin at the end of the melt season, using the latest Landsat image in the melt season (Chudley et al., 2019). The date of this Landsat image varied by year, ranging from day of year 218 (August 6, 2018) to 269 (September 26, 2016) and was typically day 250 ± 18 of the year (September 7 ±18 days). We classified lakes with less than 10% of water remaining as "completely draining", and other lakes as "partially draining". We did not classify the completeness of drainage for non-draining or uncertain 205 lakes.

Finally, we rated our classification of each lake's drainage type (fast, slow, non-draining, and unclear) and completeness (full, partial) in each year with an overall confidence level: high, moderate, and low. The confidence ratings consider the noisiness of the MODIS reflectivity ratio, the size and position of the lake within the 1×1 km box, and the presence or absence of clouds



around the lake drainage date. For instance, when observations are missing for $> 6$ days during a lake drainage, we classify the
lake as slow but with medium confidence in order to denote the possibility that it may have been a fast drainage.

### 2.3.3  Comparison to other Pâkitsoq lake-drainage datasets

While our method closely follows previous work (Selmes et al., 2011), it differs from the techniques that Morriss et al. (2013)
used to produce a dataset we leverage in this work. Morriss et al. (2013) used a normalized difference lake index approach,
while we used a reflectance ratio approach (Sect. 2.2.2). Figure 4a shows the comparison of lake-drainage date derived by us
and by Morriss et al. (2013) for melt years 2010 and 2011, our dates of overlap. Twenty-two out of 33 events (67%) agree to
within posted uncertainties in the datasets.

We also compared our dataset to its overlap with Williamson et al. (2018b). That study identified the drainage dates of fast-
draining and slow-draining lakes over a large area of western Greenland during the 2014 melt season by applying a dynamic
thresholding approach to the red band of the MODIS Terra satellite sensor (Williamson et al., 2017). This approach is quite
similar to our reflectance ratio approach, but Williamson et al. (2017) represent the lake as a single 250×250 m MODIS pixel,
while we average over a 4×4 pixel area (Fig. 2). The Williamson et al. (2017) algorithm is also fully automated, whereas
ours requires user decision making. We find that our dataset for 2014 detected 21 lake drainage events in common with the
Williamson et al. (2018b) data (Fig. 4b). Sixteen out of these 21 events (76%) agree to within posted uncertainties in each
dataset.

Overall, our dataset overlaps with existing lake-drainage datasets at 53 lakes over three years, or 17% of our 319 classified
lake-drainage events. Our overall 70% agreement with the Morriss et al. (2013) and Williamson et al. (2018b) datasets gives
us confidence to analyze the remaining 83% of the lake drainage observations in our dataset.

### 2.4  Identification of lake-draining moulins

We used WorldView imagery to manually locate the moulins that drained each lake, in years 2009–2019 and at locations where
imagery was available. We searched for and downloaded all WorldView images from the Enhanced View Web Hosting Service
(EVWHS), maintained by Maxar, with access facilitated by the Polar Geospatial Center. All WorldView images used had
resolution between 0.4–0.6 m. We used both color and single-band images, depending on availability, from WorldView-1, -2,
and -3 satellite sensors. We estimate that our moulin locations are correct to within 50 m, based on an informal comparison of
land-based features in WorldView and Landsat images. This differs from the posted  3.5–5 meter geolocation accuracy (Pope
et al., 2016; DigitalGlobe, 2016).

We viewed and analyzed all available WorldView images using QGIS, one acquisition year at a time. We searched the
vicinity of all of our 78 study lakes for moulins, using a screen scale of 1:3000, following Andrews (2015). We identified
moulins at stream termination points generally characterized by a linear fracture, a visible round hole, or both (Andrews, 2015;
Smith et al., 2015). Figure 5 illustrates the moulins we identified at Lake #31 (69.585°N, 49.600°W) from Morriss et al. (2013)
as an example.





Due to irregular image coverage in time and space, we were not able to identify the location of the moulin associated with each draining lake in every year. For 78 lakes over the 11 years 2009–2019, we identified 262 moulins associated with 66 distinct supraglacial lakes. We were able to identify moulins associated with all types of lake drainages: fast and slow; complete and partial; and high-, medium-, and low-confidence.

## 245 3 Results

### 3.1 Winter strain rates at stream-fed and lake-draining moulins

We compared the wintertime principal strain rates at moulin locations to those at all non-moulin locations in the Pâkitsoq area. We subdivided the moulins into two distinct populations: moulins that drain supraglacial lakes (Morriss et al., 2013) and moulins that are fed by supraglacial rivers, independent of lakes (Andrews, 2015). We averaged the location of each
lake-draining moulin across all years with WorldView observations.

Figure 3 shows the results of our analysis of the 20-year average wintertime principal strain rates (Joughin et al., 2016a, b) in Pâkitsoq. We distinguished the region bounding the locations of the lake-draining moulins from that bounding the stream-fed moulins (Fig. 3a and 2b). We analyzed the principal strain rates at the mean locations, over 2009–2019, of the moulins that drained 66 lakes (Fig. 3c) and at the locations of the 232 stream-fed moulins (Fig. 3d) for a total of 298 moulins. We found
that moulins are located in all strain-rate regimes: both extensional ($|\dot{\epsilon}_{1M}| < |\dot{\epsilon}_{3M}|$; $N$ = 89 of 298 moulins, or 30%) and compressional ($|\dot{\epsilon}_{1M}| > |\dot{\epsilon}_{3M}|$; $N$ = 209 of 298 moulins, or 70%). Lake-draining moulins were more often located in compressional areas (49 of 66 moulins, or 74%) than were stream-fed moulins (160 of 232 moulins, or 69%), but not significantly ($p = 0.4$). Conversely, stream-fed moulins more often occupied extensional areas (31%) than did lake-draining moulins (26%), with the same lack of significance. We compared strain rates at lake-draining moulins to the regional background. We found
that lake-draining moulins had lower more-compressional principal strain rates $\dot{\epsilon}_1$, -0.0086 yr$^{-1}$, than the regional average, -0.0053 yr$^{-1}$ ($p < 10^{-4}$). The more-extensional principal strain rate $\dot{\epsilon}_3$ at lake-draining moulins, +0.0036 yr$^{-1}$, was also lower (less extensional) than the regional average, +0.0066 yr$^{-1}$ ($p = 0.02$). These distributions are shown in Fig. 3c. Put simply, lake-draining moulins are generally located in compressional areas.

We also compared strain rates at stream-fed moulins to the regional background. We found that the more-compressional
principal strain rates $\dot{\epsilon}_1$ at stream-fed moulins, -0.0098 yr$^{-1}$, were more compressive than the regional average, -0.0083 yr$^{-1}$ ($p = 0.005$). The more-extensional principal strain rates e3 at stream-fed moulins, +0.0066 yr$^{-1}$, were significantly lower (less extensional) than the regional average, +0.0095 yr$^{-1}$ ($p < 10^{-4}$), as shown in Fig. 1d. Put simply, stream-fed moulins are also generally located in compressional areas.

Overall, we conclude with high confidence that moulins in our study area are sited in more compressive regimes than the
regional average. This conclusion is robust across moulin type (lake-draining or stream-fed) and principal strain rate ($\dot{\epsilon}_1$ or $\dot{\epsilon}_3$) but is weakest for the more-extensional principal strain rate, $\dot{\epsilon}_3$, of lake-draining moulins.



## 3.2 Moulin – lake distance

We located the moulins that drained the supraglacial lakes analyzed by Morriss et al. (2013) each year, using 2009–2019 WorldView images. Due to inconsistent coverage, we were not able to identify the moulin location for every lake in every year;
for 78 lakes over 7 years, we identified 262 moulins associated with 66 lake locations. The moulin history for Lake #31 is shown in Fig. 5. We chose this lake as an example due to the exceptionally good WorldView coverage: 7 out of 11 of the years have cloud-free scenes where we were able to visually locate the lake-draining moulin. We see multi-year reuse of the Lake #31 moulin as it advects downstream; for example, the lake drained into this moulin in 2009, 2010, and 2012, when it was respectively located 340, 350, and 520 m from the lake center (Fig. 5b–d). The ∼50-meter geolocation uncertainty associated
with WorldView images may explain variations from the expected ∼40 m/yr movement downstream based on long-term average ice velocity (Joughin et al., 2016a, a). We have no imagery from 2013, but in 2014 a new moulin formed, likely during the fast lake-drainage event that occurred between June 23–29, just 210 m from the lake center, and advected downstream to 330 m (2015) and 570 m (2016) from the lake center (Fig. 5e–f). In 2017, the lake drained rapidly through the same drainage channel and likely to the same moulin, but snow drifts obscured the view of the full path of the water (Fig. 5g).

We found that 210 (80%) of the 262 lake-draining moulins were located within 1 km of the centers of their respective lakes. The greatest observed distance between a lake and its moulin was 3.8 km, where in 2016 a slow-draining lake at ∼1010 m elevation (Lake #36) drained into a moulin at ∼980 m elevation. That moulin was associated with the rapid drainage of a closer lake (Lake #42), 400 m from the moulin, as well as a second slow-draining lake (Lake #41) located 2.1 km upstream of the moulin. Slow-draining lakes are typically associated with water overtopping the basin and flowing downstream into a moulin
(Tedesco et al., 2013; Kingslake et al., 2015). Thus, the timing of slow lake drainage reveals more about local topography and runoff rates than local stress or strain rate changes, and for that reason, we did not analyze the timing of slow-draining lakes.

Our 2009–2019 dataset of 262 lake-draining moulins contained moulins associated with 95 fast-draining lake events. The mean and median distance between fast-draining lake centers and their moulins were 463 m and 298 m, respectively (Fig. 5i). The maximum observed distance was 2.9 km, in 2010 at an elevation of ∼1200 m (Lake #18).

## 3.3 Lake drainage type

We analyzed our dataset of 78 lakes over the 11 summers (2006–2010 and 2013–2018) that followed winters with Greenland Ice Sheet velocity mosaics (Fig. 1). We sorted the lakes into four drainage types each year: Fast, slow, non-draining, and unknown. We classified all lakes as one of these four types in 2010–2019, but for 2006–2009, we use data from Morriss et al. (2013), who only identified fast-draining lakes. We continued with analysis for only lakes whose type we identified with high
confidence and all fast-draining lakes identified by Morriss et al. (2013), for a total of 319 lakes. This is a subset of our full dataset, which extends over all summers 2010–2019 and contains 356 high-confidence lake-drainage events. These extend the Morriss et al. (2013) record of 238 fast-draining lakes. The two datasets have 22 overlapping events, for a total of 527 high-confidence fast-draining lakes over the 18-year period 2002–2019.



We identified 212 high-confidence fast-draining lakes, or 25% of the theoretical maximum number of observations of all
78 lakes over the 11 summers 2006–2010 and 2013–2018. We identified 75 high-confidence slow-draining lakes (9%) and 32 high-confidence non-draining lakes (4%) over these summers. We classified the remaining 539 instances (63%) at medium or low confidence and did not analyze them further.

Our 25% fraction of fast-draining lakes is consistent with the findings of previous studies in western Greenland: 13% (Selmes et al., 2013), 30% (Morriss et al., 2013), 28% (Fitzpatrick et al., 2014), 21% (Williamson et al., 2017), 22% (Miles et al., 2017),
and 27% (Williamson et al., 2018a). One difference among these studies is our definition of fast lake drainage as $\leq 6$ days (Morriss et al., 2013), compared to $\leq 4$ days (Fitzpatrick et al., 2014; Williamson et al., 2017; Miles et al., 2017; Williamson et al., 2018a) or $\leq 2$ days (Selmes et al., 2013; Koziol et al., 2017). We reiterate that ours is not a comprehensive survey of all possible lakes, but merely a continuation of the record begun by Morriss et al. (2013) of 78 specific lakes in the Pâkitsoq region. Given our moderate number of lakes and our relatively long study period, our conclusions should be broadly applicable
to the general behavior of supraglacial lakes in slower-moving areas like Pâkitsoq.

### 3.3.1   Effect of elevation on lake drainage type

Figure 6 shows the distributions of high-confidence fast-, slow-, and non-draining lakes across elevation and strain rate regime. Because it shows only lakes whose drainage types we identified with high confidence, Fig. 6 does not accurately represent population sizes (Sect. 3.3); however, it does provide a means for comparison of elevation and strain rates across lake-drainage
type. We find that fast-draining lakes ($N = 212$) are located at lower elevations than slow- or non-draining lakes ($p < 10^{-11}$), and non-draining lakes ($N = 32$) are located at higher elevations than fast- or slow-draining lakes ($p < 10^{-12}$), as shown in Fig. 6a–b.

### 3.3.2   Effect of wintertime strain rate on lake drainage type

The lake drainage type also varies with winter principal strain rates at the lake-draining moulin. The variation of the less-
extensive winter principal strain rate, $\dot{\epsilon}_1$, with lake drainage type is shown in Fig. 6c–d. The $\dot{\epsilon}_1$ at fast-draining and at non-draining lakes was higher (more extensional) than at lake-draining moulins ($p = 0.04$ and $p = 0.007$, respectively). Though significant, we do not further analyze this difference, as lake drainage fundamentally requires extension. Thus, we concentrate on the more-extensional winter principal strain rate, $\dot{\epsilon}_3$. The variation of $\dot{\epsilon}_3$ with lake drainage type is shown in Fig. 5e–f. We find that moulins associated with fast-draining lakes ($N = 212$) are sited in more extensional areas compared to slow or non-
draining lakes ($p = 0.01$), but with a substantially weaker relationship than that with moulin elevation, as seen in the staircase plots (Fig. 6b and f). Slow-draining and non-draining lakes did not have significantly different background $\dot{\epsilon}_3$ than fast-draining lakes.





### 3.4 Lake drainage completeness

We classified each high-confidence lake drainage event as "complete" or "partial", following Chudley et al. (2019). Over the
11 summers (2006–2010 and 2013–2018) for which we have wintertime strain rate data, we found 73 complete fast-draining
events at 35 unique lakes, 16 partial fast-draining events at 13 unique lakes, 31 complete slow-draining events at 25 unique
lakes, and 42 partial slow-draining events at 27 unique lakes. This subset of our drainage data spans 60 unique lakes.

#### 3.4.1 Effect of elevation on lake drainage completeness

Figure 7 shows the distributions of complete and partial high-confidence fast- and slow-draining lakes across elevation and
strain rate regime. We find that complete fast-draining events ($N = 73$) drain into moulins at significantly lower elevations
than the other types ($p < 10^{-4}$), and that partial slow-draining events ($N = 42$) drain into moulins at significantly higher
elevations ($p < 10^{-9}$), as shown in Fig. 7a–b. Complete slow-drainage events and partial fast-drainage events did not occur at
significantly different moulin elevations.

#### 3.4.2 Effect of wintertime strain rate on lake drainage completeness

We found no significant variation of more-compressive wintertime principal strain rate $\dot{\epsilon}_1$ across lake drainage types ($p > 0.1$,
Fig. 7c–d). Complete fast-drainage events, however, were associated with moulins located in significantly more-extensional
background $\dot{\epsilon}_3$ regimes ($p < 0.003$) compared to the other types. Similarly, partial slow-drainage events were associated with
moulins located in significantly less-extensional background $\dot{\epsilon}_3$ regimes ($p < 0.003$), as shown in Fig. 7e–f. This relationship
is substantially weaker than the relationship with moulin elevation, as the relative tightness of the staircase plots (Fig. 7b and
f) demonstrate.

### 3.5 Lake drainage dates

We examined our dataset of high-confidence fast-draining lakes ($N = 212$) to find relationships between the date of fast-lake
drainage and wintertime strain rates at the lake-draining moulin, moulin elevation, or the date of melt onset that year.

#### 3.5.1 Effect of wintertime strain rate on lake drainage date

Available data allows for 11 years of comparisons of lake drainage dates (summers 2006–2010 and 2013–2018) to background
strain rates calculated from annual winter velocity mosaics (winters beginning 2005–2009 and 2012–2017). Figure 8 shows the
comparison of local wintertime strain rates to lake drainage dates the following summer. We tested for correlations between the
drainage date of high-confidence, fast-draining lakes and the principal strain rates at the site of the moulin connected to each
lake in each year ($N = 212$). We compared the drainage date to both $\dot{\epsilon}_1$ (more-compressive principal strain rate) and $\dot{\epsilon}_3$ (more-
extensional principal strain rate) at each lake-draining moulin. Within each melt season, there is a weak trend for lakes connected
to moulins sited in more-extensional regimes to drain earlier. Both principal strain rates, $\dot{\epsilon}_1$ and $\dot{\epsilon}_3$, correlate inversely with
drainage date, although neither is statistically significant in any of the 11 melt seasons we studied.





### 3.5.2 Effect of elevation and runoff on lake drainage date

We also compared the drainage dates of high-confidence, fast-draining lakes ($N = 212$) to the surface elevation of the moulin

and the date of melt onset for each year. We defined the date of first melt as the first day when the 2-meter air temperature at the Swiss Camp meteorological station exceeded zero, averaged over the four available sensors there (Steffen et al., 1996). We simultaneously regressed moulin elevation, melt onset date, winter $\dot{\epsilon}_1$, and winter $\dot{\epsilon}_3$ onto the dates of high-confidence fast lake drainage over the 11-year study interval. We found that only moulin elevation ($p < 10^{-9}$) and date of melt onset ($p < 10^{-8}$) correlated significantly. Winter strain rates $\dot{\epsilon}_1$ ($p = 0.08$) and $\dot{\epsilon}_3$ ($p = 0.62$) showed insignificant correlation with lake-drainage

date. These higher $p$-values indicate that much of the univariate correlation of $\dot{\epsilon}_1$ and $\dot{\epsilon}_3$ with drainage date, shown in Fig. 9 and presented in Sect. 3.5.1, is due to relationships between $\dot{\epsilon}_1$ and $\dot{\epsilon}_3$ and moulin elevation, most likely due to the inverse correlation of the magnitude of principal strain rates with elevation on the ice sheet (Poinar et al., 2015). Overall, our findings suggest that wintertime strain rate is not a useful predictor of lake drainage date the following summer.

### 3.6 Evolution of strain rates at GPS station-pair midpoints

We next calculate principal strain rates from the 2011–2012 GPS network data from Andrews et al. (2014, 2018). These data were recorded at 15-second intervals at eleven GPS stations separated by 2 to 20 km within a local network. Thus, these data are temporally precise, but spatially coarse (Table 1). We focused on data from the 2011 melt season.

Three nearby lakes (Lakes #52, 55, 64) drained rapidly on Day $181.6 \pm 1$ of 2011 (Morriss et al., 2013), changing the regional stress and strain rate patterns (Hoffman et al., 2018; Andrews et al., 2018). Our lake-drainage identification data

corroborate these results: we find fast drainage for Lakes #52 (days 180–181) and #55 (days 180–182) but classify Lake #64 as slow-draining (days 181–188). Detailed analysis of temporal variations in ice flow recorded by this GPS network (Andrews et al., 2018) also suggest that Lakes #52 and/or #55 drained rapidly in the evening of June 30 (day 181).

We use Equations 1–3 to calculate the horizontal principal strain rates, $\dot{\epsilon}_1$ and $\dot{\epsilon}_3$, averaged across a distance of 3.8 km separating a station pair that sits approximately along a subglacial hydropotential flowpath (Andrews et al., 2018). The upstream

station, 25N1, sits ~800 m above sea level and the lower station, FOXX, is ~700 m above sea level. Lakes #55 and #52 are respectively 12 km and 10 km upstream from the midpoint between these stations. Figure 10 shows the evolution of the local strain rates across days 180–184 (June 29 through July 3), 2011 at 15-minute, 24-hour, and 16-day resolution. Median strain rates were ~ $\pm 0.001$ yr$^{-1}$, and GPS location errors (~0.005 m per observation) manifest as a background uncertainty of 0.004 yr$^{-1}$ (Equation 5). At approximately 12:00 pm UTC (10:00 am local time) on July 1 (day 182), extensional principal strain

rates (Fig. 10a) between these two stations began to increase rapidly. They reached their peak of +0.015 yr$^{-1}$, an order of magnitude above background, four hours later and declined back to the +0.001 yr$^{-1}$ background over the following seven hours. The principal strain rate exceeded the rough threshold for crevassing, +0.005 yr$^{-1}$, for six hours; thus, new crevasses along this 3.8-km reach are likely to have opened (Hoffman et al., 2018; Stock, 2020). The compressional principal strain rate (Fig. 10b) showed coincident but substantially lower-magnitude change over this interval, suggesting that substantial vertical

strain also occurred (Andrews et al., 2018).





We smooth the GPS-derived strain rates across multiple time windows (Fig. 10). The 15-minute data resolve the transient strain rates across this eleven-hour event, while the 24-hour data show only a coarse, muted peak that barely exceeds the crevassing threshold for approximately 5 hours. The 16-day data show only a 3% change in strain rate during the event; the maximum extensional strain rate reached is 0.0012 $yr^{-1}$, essentially indistinguishable from the background.

## 3.7 Evolution of strain rates at lake-draining moulins from remote sensing data

We analyzed satellite-derived velocity data to investigate the time-evolution of strain rates at moulins across the entire Pâkitsoq region. More particularly, we studied the time evolution of strain rates at moulins associated with fast-draining lakes using two Landsat-based ice velocity products: the Technical University of Dresden (TUD) Landsat-7-derived product (Rosenau et al., 2015) and the GoLIVE Landsat-8-derived product (Fahnestock et al., 2016), both described in Sect. 2.1.2. We differenced these velocity maps according to Equations 1–3 to obtain principal strain rate maps. Next, we interpolated the strain rate time series onto the sites of moulins associated with high-confidence fast-draining lakes for each melt season. This yielded 18 (TUD) and 144 (GoLIVE) time series for $\dot{\epsilon}_1$ and $\dot{\epsilon}_3$. We shifted these time series according to the date of lake drainage, aligning all moulin strain rates in each calendar year onto a time scale referenced to the drainage date of its lake. Finally, we stacked these strain rate time series to form heat maps. We represented the likelihood of each strain rate at every time relative to the lake drainage date as a 2D distribution. We added the image pair spacing ($\Delta t$, 16 or 32 days) and the uncertainty in lake drainage date ($\delta t$, 1–6 days) to form a uniform distribution in time with width $\Delta t + \delta t$. We represented the strain rate with a normal distribution with a standard deviation of half the measurement uncertainty $\delta\dot{\epsilon}$ (Equation 4).

### 3.7.1 Landsat-7-derived strain rates

Figure 11 shows the Landsat-7-derived strain rates at moulins connected to 18 fast-draining lake events over 2002–2015, calculated from the Technical University at Dresden velocity product (Rosenau et al., 2015), which covers the Pâkitsoq region below ~900 m elevation. We selected velocity observations with time resolution of 16–32 days, denoted by the horizontal error bars on Fig. 11a. The uncertainties in these strain rate observations are large: their average, 0.23 $yr^{-1}$, is larger than typical magnitudes of the strain rates themselves, ~0.07 $yr^{-1}$. We attempt to mitigate these high uncertainties by stacking all observations to form heat maps. The heat maps (Fig. 11b–c) and the resulting time series of the most likely principal strain rates $\dot{\epsilon}_1$ and $\dot{\epsilon}_3$ show the best representation of strain rates before, during, and after these 18 fast lake-drainage events. The mean date of lake drainage was Day 190 (July 9), with standard deviation 15 days. Presence of water on the ice-sheet surface during the melt season, roughly 30 days before to 60 days after lake drainage, interferes with feature correlation between Landsat images. This manifests as a lower number of observations in this range.

The data in Fig. 11b show a slight increase in observed $\dot{\epsilon}_3$ at the time of lake drainage. Starting about 15 days before lake drainage, more-extensional $\dot{\epsilon}_3$ strain rates increase from a background of 0.02 $yr^{-1}$, reaching a peak of 0.05 $yr^{-1}$ on the date of lake drainage through about five days after. The strain rates decay over the next five days, returning to background 0.02 $yr^{-1}$ ten days after lake drainage. Within this same time period, -15 to +10 days, the more-compressional strain rates $\dot{\epsilon}_1$ (Fig. 11c) do not meaningfully change; they stay within 0.005 $yr^{-1}$ of their background value, -0.015 $yr^{-1}$. The data do not show a





significant spring speedup event, which we would expect in May (Andrews et al., 2018), about 30 days before the average lake
drainage date on Fig. 11. The data show an unexpected feature that spans 50–70 days after lake drainage; the highest average
$\dot\epsilon_3$ in the dataset (+0.1 yr$^{-1}$) and the lowest average $\dot\epsilon_1$ (-0.04 yr$^{-1}$) appear in this range. This range spans late August through
mid-September and roughly coincides with the end of the melt season, when the subglacial hydrologic system transfers back
to an inefficient state and sudden water input events, such as rain storms, can re-accelerate ice flow and manifest as extensional
strain rate events (Doyle et al., 2015; Horgan et al., 2015).

### 3.7.2 Landsat 8-derived strain rates

Figure 12 shows the strain rates at moulins connected to 152 fast-draining lake events at 59 distinct lakes over 2013–2019,
calculated from the Global Land Ice Velocity Extraction from Landsat 8 (GoLIVE) velocity product (Fahnestock et al., 2016),
which covers the entirety of our study area. We selected velocity observations with time resolution of 16 days (horizontal
error bars on Fig. 12a). This differs from our 16–32 day choice for the TUD velocities for two reasons: (1) There were more
GoLIVE observations available overall, allowing tighter constraints in image spacing; and (2) including 32-day observations
significantly smoothed any strain rate signal (not shown). The uncertainties in the Landsat 8 GoLIVE-derived strain rates are
smaller than those in the Landsat 7 TUD-derived product, but their average (0.11 yr$^{-1}$) is still larger than typical strain rate
magnitudes ($\sim$0.05 yr$^{-1}$). The mean date of lake drainage was the same as in the 2002–2015 dataset: Day 190, or July 9, with
standard deviation 17 days.

As with the TUD-derived strain rates, the presence of water on the ice-sheet surface is a major impediment to velocity data
collection in our study area. The GoLIVE data have a data gap from $\sim$50 days before the mean lake drainage through $\sim$10
days after. Thus, although we have 288 strain rate observations at 152 sites of fast-draining lake moulins through seven melt
seasons, none of these can provide information on strain rates in the days, or even weeks, preceding fast lake drainage.

Despite the nonutility of the GoLIVE-derived strain rate data for studying lake drainage events, the data do show some
features over the rest of the melt season. The coverage gap precludes any spring speedup feature, which we would expect $\sim$30
days before lake drainage. The GoLIVE data show an end-of-melt-season feature roughly 60–80 days after lake drainage. As
with the TUD data, the peak $\dot\epsilon_3$ strain rate (Fig. 12b) occurs 70–75 days after lake drainage (+0.09 yr$^{-1}$). The $\dot\epsilon_1$ strain rate
(Fig. 12c) shows little variation throughout the time series, generally remaining within 0.01 yr$^{-1}$ of its -0.014 yr$^{-1}$ mean.

## 4 Discussion

This work is motivated by the apparent contradiction that moulins, which require extension to form or activate, are often located
in or near lake basins, where ice flow is generally compressive (Alley et al., 2005; Das et al., 2008; Krawczynski et al., 2009).
This pattern suggests that strain rate transients – perturbation events on the scale of hours to days – are responsible for moulin
formation or re-activation (Stevens et al., 2015). In that conceptual model, subglacial water pulses briefly change the local
surface strain rates, in some places pushing ice above its fracture threshold to form new moulins near or within supraglacial
lake basins (Tedesco et al., 2013; Stevens et al., 2015; Christoffersen et al., 2018; Hoffman et al., 2018; Chudley et al., 2019),





or re-activating existing moulins. We investigated the degree to which elevation and background ice flow patterns influence moulin formation or re-activation, and the degree to which current remote-sensing datasets can resolve strain rate transients that may drive moulin activation.

## 4.1 Siting and timing of lake-draining moulins

### 465    4.1.1    Moulin locations

We find that on average, stream-fed and lake-draining moulins are located in more compressional regimes than the regional average, although this conclusion was weakest for the more-extensional principal strain rate, $\dot{\epsilon}_3$, of lake-draining moulins. However, $\dot{\epsilon}_3$ is a more relevant metric for fracture initiation than $\dot{\epsilon}_1$, so this finding is somewhat surprising. We explore some potential explanations below.

Figure 5 illustrates a typical feature of our dataset: lake-draining moulins can be located some hundreds to thousands of meters downstream of the lake that they drain. Meltwater is carried through supraglacial streams or deeply incised ice canyons (Joughin et al., 2013; Smith et al., 2015; Poinar et al., 2015; St Germain and Moorman, 2019). In Pâkitsoq, ice flow varies from compressional to extensional on spatial scales of roughly 5 km (Catania et al., 2008), or 4–6 ice thicknesses (Sergienko et al., 2014; Gudmundsson, 2003). Thus, the crevasses that gave rise to these moulins may have formed in more extensional regimes,
then advected into more compressive ones. The typical regional ice speed of ∼100 m/yr would require the moulins to persist for at least ten years to travel the >1–2 km required to cross into a new strain rate regime (Catania and Neumann, 2010). This is substantially greater than the observed lake-draining moulin lifetime in Pâktisoq, 2–5 years (Andrews, 2015), also illustrated in Fig. 5. Thus, there is likely not enough advection over the lifetimes of most lake-draining moulins to transfer them from an extensional zone of formation to a compressional zone of observation.

Our stream-fed moulin dataset spans lower elevations, where typical ice thickness (∼600 m) suggests an extensional-to-compressional scale of roughly 2–3 km. The slightly faster ice flow here could allow a stream-fed moulin to form in an extensional area and advect into a compressional zone within 5–10 years. Thus, advection is a potential explanation for the high fraction (69%) of stream-fed moulins located in compressional zones. It remains a less likely mechanism for the similarly high fraction (74%) of compressionally sited lake-draining moulins, which are higher on the ice sheet.

Finally, our strain rate data, derived from the long-term average velocities (Joughin et al., 2016a) and smoothed over 1 km, may obscure fine-scale variations in strain rate. This was recently demonstrated by Chudley et al. (2020), who conducted a UAV survey to measured the surface strain rates on a fast-moving outlet glacier at ∼6 m resolution. Their dataset showed a high degree of spatial variability, which the ∼200 m resolution MEaSUREs data captured accurately but coarsely. Thus, it remains possible that moulins occupy local (<200 m) extensional areas in regionally (>200 m) compressional areas.

### 490    4.1.2    Distinctions among fast-draining, completely draining, and bottom-draining lakes

Our study lakes have an average radius of 390 m (Morriss et al., 2013). Figure 5i shows that 61 of the 95 (64%) moulins associated with fast-draining lakes are bottom-draining moulins, defined as being located within 390 m of the lake center. An





additional 23 (24%) moulins are located within 1 km of lake centers, and the remaining 11 fast-draining-lake moulins (12%) are more than 1 km away. Thus, some 40% of the lakes that we classified as fast-draining are not bottom-draining.

We present a simple model consistent with our observations. Fast-draining lakes form moulins by lake-bottom hydrofracture (e.g., Das et al., 2008; Stevens et al., 2015). In subsequent years, lake water flows into that same moulin, via supraglacial streams, as it advects up to ∼1 km downstream. Fast lake drainage can continue to occur, even though the moulin is no longer located in the lake bottom, but slow lake drainage is also possible (Tedesco et al., 2013). During this period, fast but partial lake drainages are likely to occur (Chudley et al., 2019). After ∼5 years, the increasing distance of the moulin from the lake, creep closure, strain rate regime change, topography change, and/or ablation of the outflow channel, separates the moulin from the lake water, and a new lake-bottom moulin forms through hydrofracture.

The farthest moulins were located 1.7 km, 2.5 km, and 2.9 km from lake centers. These moulins were associated with fast-draining lakes upstream of other fast-draining lakes, connected by >1 km long supraglacial streams. While these lakes meet the rigid criteria for "fast-draining", they are not locally draining lakes (Poinar et al., 2015); instead, they drain by continual downward incision of an outflow channel (Banwell et al., 2012; Kingslake et al., 2015). The relative rates of channel incision and lake water-level change determines the completeness of lake drainage: a higher channel incision rate will completely drain the lake, a lower channel incision rate will only partially drain the lake (Kingslake et al., 2015). Larger, higher-elevation lakes, which tend to be shallower, are more likely to drain completely. In fact, two of our three fast-draining lakes farthest from their moulins (Lake #18 in 2010; Lake #30 in 2019) were completely draining lakes; Lake #18 is higher-elevation (1200 m) and Lake #30 is relatively large (630 m radius). The other lake (Lake #52 in 2015), has average size, average elevation, and an elongated shape; it drained partially into a moulin 1.7 km downstream.

This work, and other recent detailed analyses (e.g., Chudley et al., 2019; Williamson et al., 2018b; Morriss et al., 2013), make it clear that fast-draining, completely draining, and bottom-draining are not synonyms. The speed of drainage, completeness of drainage, and location of the lake-draining moulin are all independent dimensions of lake drainage.

### 4.1.3 Moulin elevation and lake drainage type

Our results show a distinct sorting of lake drainage speed by elevation (Fig. 6), with fast-draining lakes at lower elevations ($p < 10^{-11}$) and non-draining lakes at higher elevations ($p < 10^{-14}$). Recent work by Cooley and Christoffersen (2017) that carefully controlled for the possibility of misclassifying lake drainage type due to cloudiness biases, however, found no relationship between lake elevation and drainage type. For comparison, our MODIS dataset only contains cloud-free pixels, and our analysis requires a lake to have a "filled" observation and a "drained" observation within 6 days of each other in order to be classified as fast-draining. Thus, if a fast lake drainage occurred during a $\geq$ 6-day cloudy period, our analysis would misclassify it as a slow drainage. Cooley and Christoffersen (2017) identified this as a common shortcoming of existing lake-drainage-date datasets.

Cooley and Christoffersen (2017) applied a 6-day drainage criteria similar to ours, but over a larger study area and longer time period. They found that 22% of lakes drained rapidly each season. When they controlled for the possibility of "missed" fast drainages due to cloudy periods, they calculated that 43% of lakes would drain rapidly, and that any relationship between





lake elevation and drainage type vanished. The uncorrected results of Cooley and Christoffersen (2017) are comparable to our finding that 25% of lakes in Pâkitsoq drained rapidly, suggesting that the true ratio is higher than our result.

For insight, we look to slow-draining lakes in our dataset. We incorporated the cloud state during the lake drainage into our confidence ratings (high, medium, and low; Sect. 2.2.2). For instance, when clouds obscured observations for >6 days during a lake drainage, we classified the lake drainage as slow but assigned it medium confidence to denote the possibility that it may have been a fast drainage. Our dataset contains $N = 49$ medium-confidence slow-draining lakes. These average 98 m higher in elevation than the high-confidence fast-draining lakes. If fast-draining and slow-draining lakes in our study area were to be sited at indistinguishable elevations ($p > 0.05$), to match the results of Cooley and Christoffersen (2017), this would require that the 26 (53%) highest-elevation medium-confidence slow-draining lakes (1110 m $< s <$ 1410 m) were actually fast-draining. While possible, this seems unrealistic, as cloudiness should affect identification of slow-draining lakes all elevations (530 m $< s <$ 1410 m) equally. Instead, we randomly selected a subset of the medium-confidence slow-draining lakes and reassigned them as fast-draining. We found that the elevations of fast-draining lakes were still significantly different from slow- and non-draining lakes, with average $p = 0.008$. In fact, we were not able to match the indistinguishibility found by Cooley and Christoffersen (2017) to $p > 0.05$ in ten million sets of random selections.

Although we represented uncertainty due to cloudiness, we were unable to reproduce the findings of Cooley and Christoffersen (2017) that lake elevation is not correlated to drainage type. Our results agree with previous findings that fast-draining lakes occupy lower regions of the ice sheet (Selmes et al., 2013; Morriss et al., 2013; Miles et al., 2017).

### 4.1.4 Moulin background strain rate and lake drainage timing

There is suggestive observational evidence for lake-drainage cascades, whereby water supplied to the subglacial system by drainage of an initial lake is linked to drainage of neighboring lakes (within tens of kilometers) in the following days (Fitzpatrick et al., 2014). Modeling work supports the physical feasibility of such cascades, whether the initial lake is upstream (Christoffersen et al., 2018; Hoffman et al., 2018) or downstream (Price et al., 2008; Stock, 2020) of the cascading lakes.

We investigated whether lakes in more-extensional regimes drained first and triggered regional cascades but found that this was not likely. In some years, higher-elevation lakes were the first to drain: notably 2007, 2009, 2015 (Fig. 8), but the winter strain rates at these lakes were not different from the population average of lake-draining moulins. Background principal strain rates can influence the orientation of moulin-forming fractures in the bottoms of fast-draining lakes (Chudley et al., 2019), but we found no significant relationship to lake drainage date (Fig. 8).

Our results are consistent with those of Williamson et al. (2018b), who tested whether fast-draining lakes were statistically different from slow-draining and non-draining lakes with respect to a full suite of parameters, including winter strain rates at the ice-sheet surface. Williamson et al. (2018b), however, calculated strain rates from a velocity dataset that draws from wintertime measurements across ~20 years (Joughin et al., 2016a), which differs from our more time-specific analysis of strain rates observed in the winter immediately preceding each melt season when lake drainages were observed (Joughin et al., 2018b). Our more specific analysis also finds no statistically significant correlation between winter strain rate and lake drainage date, lending further support to the Williamson et al. (2018b) finding that fast-draining lakes do not occupy different





wintertime strain-rate regimes than other supraglacial lakes. The results also align with those of Emetc et al. (2018), who found only limited spatial correlation between Antarctic ice-shelf fractures and background (fall/winter/spring composite) strain rates from Rignot et al. (2011).

Indeed, the earliest fast-draining lake in Pâktisoq each year is not generally sited in a background extensional area, but
rather most often a lower-elevation lake (Fig. 8). Earlier access of meltwater to the bed at lower elevation is well documented (McGrath et al., 2011; Bartholomew et al., 2011; Doyle et al., 2013), but such access is often through crevasse fields or stream-fed moulins, not necessarily through moulins fed by fast-draining lakes (Catania et al., 2008). Indeed, the first fast-draining lake of each year occurred 10–60 days after the day of first runoff recorded at Swiss Camp (Fig. 8), whereas melt-induced ice flow has been observed 3–10 days after melt onset in our study area (Andrews et al., 2018). This contrast suggests that water
reaches the bed via other means before fast lake drainages begin.

### 4.2   Prediction of future lake-drainage events

#### 4.2.1   Progress in mapping current features versus identifying predictive metrics

A multitude of intensive, independent, and international efforts over the past decade has yielded multiple reliable algorithms and workflows that date and detect lake drainages from satellite imagery (e.g., McMillan et al., 2007; Sundal et al., 2009;
Selmes et al., 2011; Liang et al., 2012; Morriss et al., 2013; Johansson et al., 2013; Fitzpatrick et al., 2014; Williamson et al., 2017). A related remote-sensing analysis push has yielded techniques that produce detailed maps of lakes and streams on the ice-sheet surface, which are readily applied to locate moulins at stream terminations (Yang and Smith, 2013; Smith et al., 2015; Yang and Smith, 2016; King et al., 2016; Andrews et al., 2018; Yuan et al., 2020). These sophisticated remote-sensing observations of supraglacial lakes, streams, and moulins cover the entirety of Southwest Greenland and extend more than a
decade in time. The high data volume should have utility for identifying patterns in lake-drainage date and moulin location, as attempted in this study and by Williamson et al. (2018b). However, beyond coarse trends in lake elevation, which are disputed (Cooley and Christoffersen, 2017), runoff timing (Fig. 9), and lake size (Williamson et al., 2018b), we find no lake or local ice-sheet properties that are predictive of drainage date. Thus, despite the community's state-of-the-art near-real-time analysis capabilities of ever-increasing image data, we regard a "more data" approach to identifying predictive lake-drainage metrics
with pessimism.

#### 4.2.2   Current remote-sensing products for regional-scale strain rates

Precise measurements from GPS networks show short-lived, high-magnitude strain rate events that precede fast lake drainages (e.g., Fig. 10; Stevens et al. (2015)). Recent research has provided a mechanistic understanding that these events induce lake drainage (Tedesco et al., 2013; Stevens et al., 2015; Hoffman et al., 2018; Christoffersen et al., 2018). The GPS network data
shown in Fig. 10 are typical, however, in that their spatial resolution is limited ($\sim$4 km in our example). Notably different is the large field campaign by Stevens et al. *Stevens:2015ht, which installed and maintained 15 GPS stations within a 100 km$^2$ area around a pair of lakes across three melt seasons and successfully measured precursor strain rates in unprecedented spatial detail.





Inspired by that field effort, we consider a hypothetical regional-scale GPS network (>1000 km$^2$) with fine enough resolution to resolve strain rates in individual basins (<∼3–5 km; Fig. 10). Such a network would require dozens of GPS units, making
installation in the ablation zone a serious logistical challenge. Thus, to study the relationship between strain rate perturbations and activation times of moulins in multiple basins across a region, remote-sensing data is required.

Existing regional-scale ice-sheet velocity data products calculate ice flow across multi-day periods, compared to sub-daily flow perturbations that likely accompany fast lake drainages (Fig. 10). Here, we found that these strain rate perturbations are not resolved by velocity products derived from Landsat 7–8, despite stacking over 170 high-confidence, fast-drainage events
(Fig. 11–12). The Landsat 7 observations hint at a strain rate peak when lakes drain, but the signal is ambiguous and has little hope for predictive power. The Landsat 8 observations are especially inhibited by surface water, which interferes with velocity observations during the peak of the melt season, when most lakes tend to drain. Thus, we cannot identify relationships between lake drainage dates and seasonal strain rates from currently available regional velocity datasets. Other imagery, such as high-resolution WorldView or Planet products, for example, generally have poorer georeferencing, making ice-flow velocity
extraction challenging (Armstrong et al., 2016).

### 4.2.3 Future approaches to frequent measurements of regional-scale strain rates

**Sentinel-1 velocity products**

Our study did not investigate use of velocity products derived from Sentinel-1 data, which are newly becoming available. The current family of products applies intensity tracking to C-band SAR images from the Sentinel-1a/b satellite pair to calculate
ice surface velocities (Lemos et al., 2018b). Full-Greenland maps at 24-day intervals with 500 m resolution are publicly available (Solgaard et al., 2018). Velocity maps for a number of outlet glaciers in Greenland with 6- to 12-day spacing and 100–400 m resolution are being produced by multiple groups (Wuite et al., 2016; Hogg et al., 2015; Lemos et al., 2018a), but products for our study area are not currently available. Planned launch of Sentinel 1c/d satellites, in 2022–2023 or beyond, could bring this interval down to 1–2 days if Sentinel 1a/b outlive their expected times of service. The NASA-Indian Space
Agency SAR (NISAR) mission, planned for 2021, will add further ice-sheet velocity measurements at 12-day intervals (Joughin et al., 2018a). However, as with the Landsat-derived velocity products, the presence of water on the ice-sheet surface causes decorrelation between images, reducing the number and spatial coverage of velocity observations during the melt season. Overall, resolving short-duration events in the vicinity of supraglacial lakes, streams, and rapidly changing surface conditions may remain out of reach for even the multitude of forthcoming SAR-based products that will provide unprecedented temporal
resolution of regional ice-flow observations.

Table 1 and Fig. 13 detail the spatial and temporal resolution and coverage of the various methods to observe surface strain rates that we address here. Table 1 and Fig. 13 also tabulate the characteristics of an ideal method for observing rapid strain rate perturbations associated with fast lake drainage. This ideal observation method has four important traits: (1) Regional spatial coverage, spanning >∼30 km along a transect in Pâkitsoq, for instance, in order to sample a range of elevations and lake types.
(2) Moderately high spatial resolution. Because calculation of strain rates requires averaging over >∼1 ice thickness (∼1 km in Pâkitsoq), spatial resolution ≤∼ 500 m is needed. While the pixel resolution of the MODIS sensors is 250–500 m, image



correlation requirements of at least 15×15 pixels (e.g., Rosenau et al., 2015) would reduce the resolution of a hypothetical MODIS velocity product by an order of magnitude or more, to >4 km. (3) High temporal resolution, with the ability to resolve daily to hourly fluctuations (Fig. 10). This occupies a middle ground between GPS units, which typically record every 5–30
seconds during months with insolation, and satellite-based observations, which have repeat passes every 16 days or more. (4) Ability to resolve ice flow regularly during the melt season, even in the presence of water on the ice surface. Overall, this ideal observation method does not currently exist.

**Photogrammetry observations**

Unmanned aerial vehicles (UAVs) are rapidly becoming a standard field tool. UAV-based photogrammetry can supply high-
resolution strain rate maps with daily to sub-daily spacing. For instance, Chudley et al. (2019) produced two 6.4-m-resolution velocity maps of a 0.3 km$^2$ area by four UAV flights in a single field season. Velocity maps of a calving front by Ryan et al. (2015) achieve uncertainty of ∼10%. However, temporal coverage of this tool is limited to the duration of field camp occupation by the research group, and spatial coverage is limited to the range of the UAV, which is typically on the scale of 1–2 lake basins. Together, these limit the applicability of UAV photogrammetry to our problem (Fig. 13).

Photogrammetry by time-lapse camera holds promise, especially for glacier fronts monitored by rock-mounted cameras over multi-year periods. For example, Ahn and Box (2010) derived daily velocity maps of four fast-moving Greenland outlet glaciers using the Extreme Ice Survey megapixel cameras. Measuring ice flow slower than ∼1 m/day, however, which is typical of Pâkitsoq, may be less feasible (How et al., 2019). Furthermore, time-lapse photogrammetry requires nunataks or fjord walls to mount stationary cameras, so its application in broad icy areas that lack nunataks, such as the extensive ablation zone across
western Greenland, would be limited.

Photogrammetry is also possible from repeat airline flights and can outperform photogrammetry by fjord-wall-mounted cameras (Eiken and Sund, 2012). This was studied on Nathorstbreen, Svalbard, by Eiken and Sund (2012), who worked with pilots who regularly overflew the glacier at ∼20,000 feet on a commercial airline route. The pilots snapped photographs of the glacier from the cockpits of Boeing 373s using off-the-shelf, personally owned cameras in 2009. Eiken and Sund (2012) then
derived velocity maps of the glacier terminus from the photographic data. The proximity of the Southwest Greenland ablation zone to the Kangerlussuaq Air Base, with its near-daily Air Greenland Airbus A330 round-trip flights to Copenhagen, could allow similar production of long-term, frequent-duration, high-resolution velocity maps with regional coverage (Fig. 13).

**Dense regional GPS networks**

Previous work has used GPS networks to study precursor drainage events at individual lakes (Doyle et al., 2013; Stevens
et al., 2015; Hoffman et al., 2018). Given the shortcomings of the satellite-based data sources discussed above, GPS networks are likely the best option for resolving surface strain rates at fine time scales and sub-regional spatial scales (Fig. 13, Table 1). However, the number of stations needed to study an entire region, such as Pâkitsoq, would introduce serious logistical challenges. A station density of 1–2 stations per 10 km span is likely sufficient to detect precursor events or their absence (Fig. 10); thus, covering the Pâkitsoq study area would require ∼20–50 stations. This number is much higher than typically
managed. Furthermore, to characterize the magnitude of the strain rate precursors, a spatial density approaching 1 station per





km, as used by Stevens et al. (2015), is likely necessary. This would increase the number of stations required by an order of magnitude. Design of any such GPS network will require careful consideration of the trade-offs in spatial resolution, spatial coverage, and the cost and feasibility to install and maintain stations in the fidgety conditions of ice-sheet ablation zones.

### 4.3   Parameterizing moulins in ice-sheet models

Ice-sheet models, which are based around detailed physical representation of ice flow, are central tools for state-of-the-art projections of ice-sheet mass balance (Nowicki et al., 2016, 2020). Societal ability to accurately predict global sea levels in future climates depends, in part, on the fidelity of ice flow patterns predicted by these models. This, in turn, depends on sliding speeds (Rückamp et al., 2020) via basal hydrologic conditions, which evolve according to the spatial distribution and timing of melt water delivery to the subglacial system (e.g., Banwell et al., 2016). State-of-the-art ice-sheet projections, such

as those from ISMIP6, currently do not explicitly include meltwater inputs to the bed (Nowicki et al., 2020), and most current development work in this area is focused on climate model – ice-sheet model coupling for improving surface mass balance (e.g., Goelzer et al., 2020). Focused development on linking runoff to ice flow via meltwater inputs to the bed is likely multiple years in the future.

Thus, we look to models of subglacial hydrology, which contain the most detailed representation of moulins in predictive

models to date. To deliver surface melt at specific points and times to their subglacial model, Banwell et al. (2016) used a DEM to find local minima, then used a surface water routing and lake-filling scheme to trigger moulin formation at each topographic minimum once a meltwater accumulation threshold was reached. Similarly, de Fleurian et al. (2016) identified moulin locations as stream terminations a Landsat 8 image, used a DEM to quantify the size of the catchment of each moulin, and input time-dependent, RACMO-derived meltwater fluxes within the catchment through each moulin to the bed in real

time. Finally, Clason et al. (2012, 2015) estimated crevasse depths and locations using ice velocities, a stress threshold, and meltwater accumulation data. They forecasted the date of meltwater input to the bed as the day the crevasse depth reached the full ice thickness. Overall, existing methods to locate moulins in space are largely reliant on present-day observations, and some current methods to construct time-varying basal input hydrographs ignore the episodic nature of moulin activation. Thus, we currently lack a method to accurately represent moulin locations and hydrographs in models of ice sheets in future climates.

We propose a potential path forward for the spatial and temporal representation of moulins in ice-sheet models. First, we look at spatial forecasting. At low elevations, surface-to-bed connections are ubiquitous via stream-fed moulins (Fig. 3) and crevasse fields (McGrath et al., 2011). At high elevations, most moulins are associated with lakes (Smith et al., 2015; Yang and Smith, 2016). Lake locations are static (Lampkin and VanderBerg, 2011), and near-future lake locations, in topographic lows that are currently above typical elevations with runoff, are predictable from current DEMs (Leeson et al., 2014; Ignéczi

et al., 2016). Finally, locations of lakes in the distant future, or in any scenario in which ice-sheet size and geometry are substantially different from today, are predictable in ice-sheet models run with sufficient resolution (<∼1 km) to resolve local topographic depressions. If lake locations are forecastable, what remains is predicting whether the lakes will drain to local moulins or to moulins far downglacier, via commonly observed long outflow streams (Poinar et al., 2015). For this, we look to Stock (2020), who used an efficient transfer-function model (Gudmundsson, 2003, 2008) to quantify the probability of moulin





formation induced by basal water perturbations (e.g., Stevens et al., 2015; Christoffersen et al., 2018). The Stock (2020) model is applicable to arbitrary ice-sheet configuration, so it could be used to generate moulin locations in future ice-sheet conditions.

Next, we look toward forecasting the date of activation of an arbitrary moulin. Important metrics are elevation on the ice-sheet surface and cumulative runoff, although together these explain less than half of the variance in the date of fast lake drainage (Sect. 3.5.2). The remaining variance in moulin activation date may lie in strain-rate precursors (Christoffersen et al.,

2018) instigated by variations in basal hydrology. Resolving these precursors is within reach of future coupled subglacial hydrology – ice sheet models, although they will require fine spatial ($<\sim$1 km) and temporal ($<\sim$1 day) resolution. Elevation, runoff, and time varying strain rates are potential predictor variables that are readily available within ice-sheet models and their climate forcings; thus, there is potential to construct accurate basal water input hydrographs even in future ice-sheet states.

The factors that determine the spatio-temporal distribution of moulins are finer than spatial and temporal resolution of most

ice-sheet-scale models. Thus, we anticipate that some parameterization will be needed to achieve computational feasibility; we envision a probabilistic or ensemble approach to representing meltwater delivery to the bed. For instance, in the absence of discrete moulin locations, moulin density may suffice (Banwell et al., 2016). The spatial density of moulins could be approximated using bedrock geometry, ice-sheet thickness, and slip ratio (Stock, 2020), all of which are common ice-sheet-model variables, as well as less well represented parameters related to local basal water perturbations throughout a melt season.

These hydrological quantities could potentially be calibrated using currently available image- or field-based surveys of moulin locations, then applied to future climate scenarios and ice-sheet geometries. Ultimately, a probabilistic map of moulin location and activation date within the melt season could be supplied to subglacial hydrology models. In time, coupled ice sheet – subglacial hydrology models, driven with the parameterized basal input hydrographs we envision, could supply better-informed projections of ice flow variability in future climates.

# 5 Conclusions

In this study of 78 lakes in the Pâkitsoq area of western Greenland over 18 melt seasons, we identified 527 lake-drainage events with high confidence. We found that lake elevation and date of melt onset, but not the annual-average strain rate, were meaningful predictors of the timing of fast lake drainage ($N = 212$). We also investigated the hypothesis that transient, high-magnitude strain rate events regularly precede fast lake drainages, but fundamental issues with data coverage and temporal

resolution limited our ability to make significant conclusions.

Our study is motivated by ongoing rapid development and application of ice-sheet models that forecast likely states of the Greenland and Antarctic Ice Sheets in future climate scenarios. Especially for Greenland, the coming generation of ice-sheet models will need to parameterize the timing and locations of melt delivery to the bed. State variables like ice thickness and strain rates are attractive potential parameters because these are available at every model time step. However, our finding that

background strain rates do not significantly correlate with lake drainage date implies that this is not likely to provide a path forward. Better metrics for predicting lake-drainage dates are elevation on the ice-sheet surface (lakes associated with lower moulins drain first), and meteorological data such as the date of surface melt onset (lakes drain earlier in years with early melt).



Whether our second hypothesis, that precursor strain-rate perturbations precede lake drainage, will be able to be evaluated on the regional scale remains unclear. Currently available remote-sensing velocity data products for Pâkitsoq calculate ice flow across multi-day periods ($\geq 16$ days) and thus cannot resolve the sub-daily flow perturbations that are hypothesized to incite fast lake drainage. Forthcoming Sentinel-1 velocity maps will improve the temporal resolution to $\geq 6$ days, and the planned launch of the Sentinel 1c and 1d satellites could lower this interval to 1–2 days, but even at this unprecedented temporal resolution of regional ice-flow observations, ability to resolve sub-daily ice flow transients is not certain. This limitation stems from the prevalence of water on the ice-sheet surface, which severely interferes with signal extraction and will most likely limit the time resolution to multiple days.

We conclude that field methods are likely required to measure strain rates at sufficient temporal resolution to test the precursor hypothesis. Developing techniques that can operate at the regional scale, frequently, and despite the presence of water, may be required. Candidate methods include airborne or in-situ photogrammetry, although challenges remain in capturing ice flow in areas far from nunataks, or vast GPS networks, despite their cost and logistical burden at a regional scale.

We require a fuller understanding of the role of high-magnitude, short-duration strain-rate events in fast lake drainage, including the magnitude of these perturbations and how often fast lake drainages occur in their absence, if we are to parameterize moulin timing for use in the basal hydrological components of ice-sheet models.

*Data availability.* The 2009–2019 lake drainage date dataset, the 2009–2019 moulin locations, and all strain rate maps used in this study are available through the University at Buffalo Libraries at http://hdl.handle.net/10477/82127. The GPS data are archived as indicated in Andrews et al. (2018).

*Author contributions.* K.P. and L.C.A. conceived the research ideas together. K.P. designed the study, obtained the data, performed the analyses, created all figures, and wrote the manuscript. L.C.A. contributed ideas, discussion, and manuscript editing.

*Competing interests.* The authors declare that they have no conflict of interest.

*Acknowledgements.* The MODIS data products (MOD09GQ, MOD09GA, MYD09GQ, and MYD09GA) were retrieved from the online Data Pool, courtesy of the NASA Land Processes Distributed Active Archive Center (LP DAAC), USGS/Earth Resources Observation and Science (EROS) Center, Sioux Falls, South Dakota, via Google Earth Engine. We acknowledge DigitalGlobe, Inc. for providing WorldView images. K.P. was supported by the NASA Postdoctoral Program (NNH15CO48B), NASA Cryosphere grant 80NSSC19K0054, and the Research and Education in eNergy, Environment and Water (RENEW) Institute at the University at Buffalo. L.C.A. acknowledges support from NASA Cryosphere grant 80NSSC19K0054 and from the Global Modeling and Assimilation Office at NASA Goddard Space Flight





Center funded under the NASA Modeling, Analysis, and Prediction (MAP) program. We thank Sophie Nowicki for helpful discussions over the lifetime of this project.



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



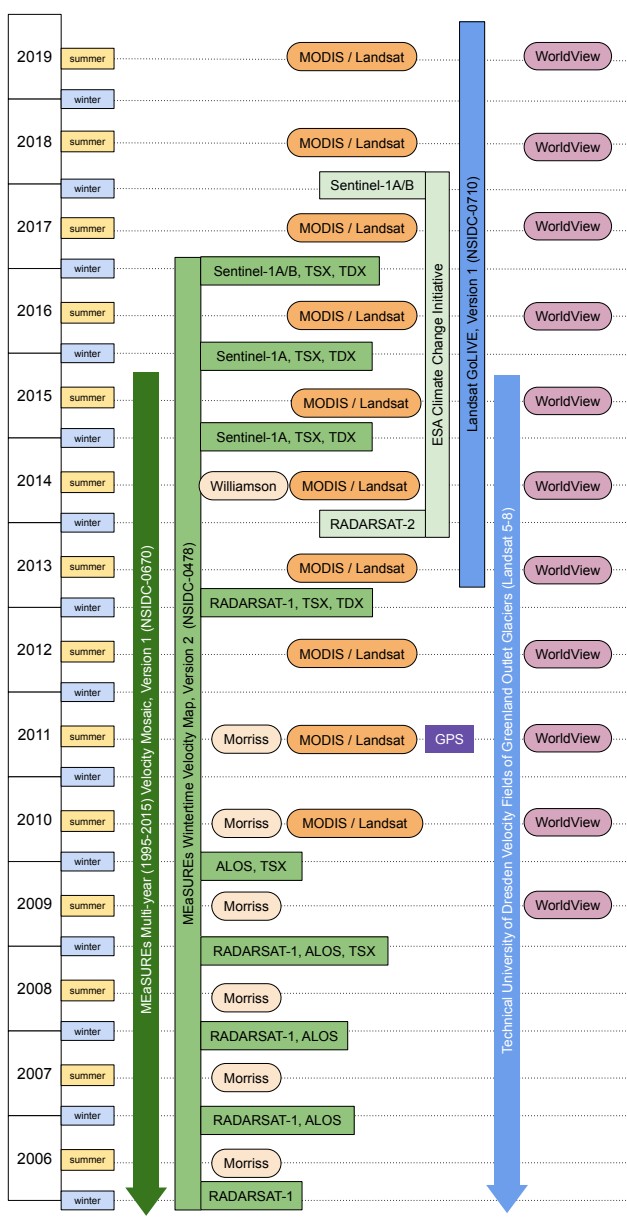

**Figure 1.** Datasets used in this analysis, organized by year and season. We calculate strain rates from SAR-based velocity mosaics (green; data provider and satellite sources noted), which are based on wintertime data, as well as from optically based velocity mosaics (blue), which are available during seasons with solar illumination. Lake drainage data sources are shown in orange; Morriss refers to Morriss et al. (2013), and Williamson refers to Williamson et al. (2018b). The Morriss et al. (2013) record begins in 2002 (not shown). We overlap with one complete field season (2011) of GPS-derived strain rates (purple), from Andrews et al. (2014, 2018). We use high-resolution, July–October WorldView images (light purple), when available, to locate the moulins that drain the lakes in our study area each year.


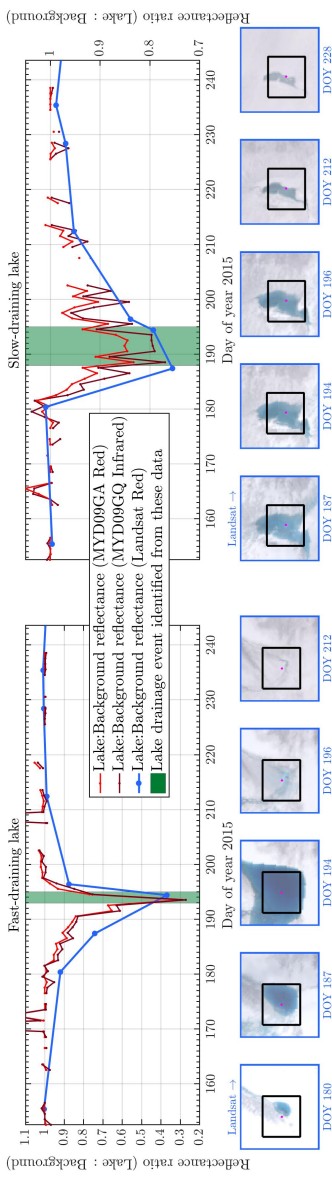

**Figure 2.** Illustration of lake drainage detection datasets on an example (a) fast-draining lake and (b) slow-draining lake in the year 2015. We use the techniques of Selmes et al. (2011) to compare the red (MODIS, Landsat) and infrared (MODIS) reflections from a 1 km×1 km box centered on each lake (black box in lower panels) to the 2 km×2 km box surrounding each lake (lower panels). We identify the day of lake drainage as the date range when this reflectance ratio increases from its minimum value (green bar). This method allows us to qualitatively distinguish fast lake-drainage events from slow events and to quantify the date of drainage, including an estimated uncertainty.





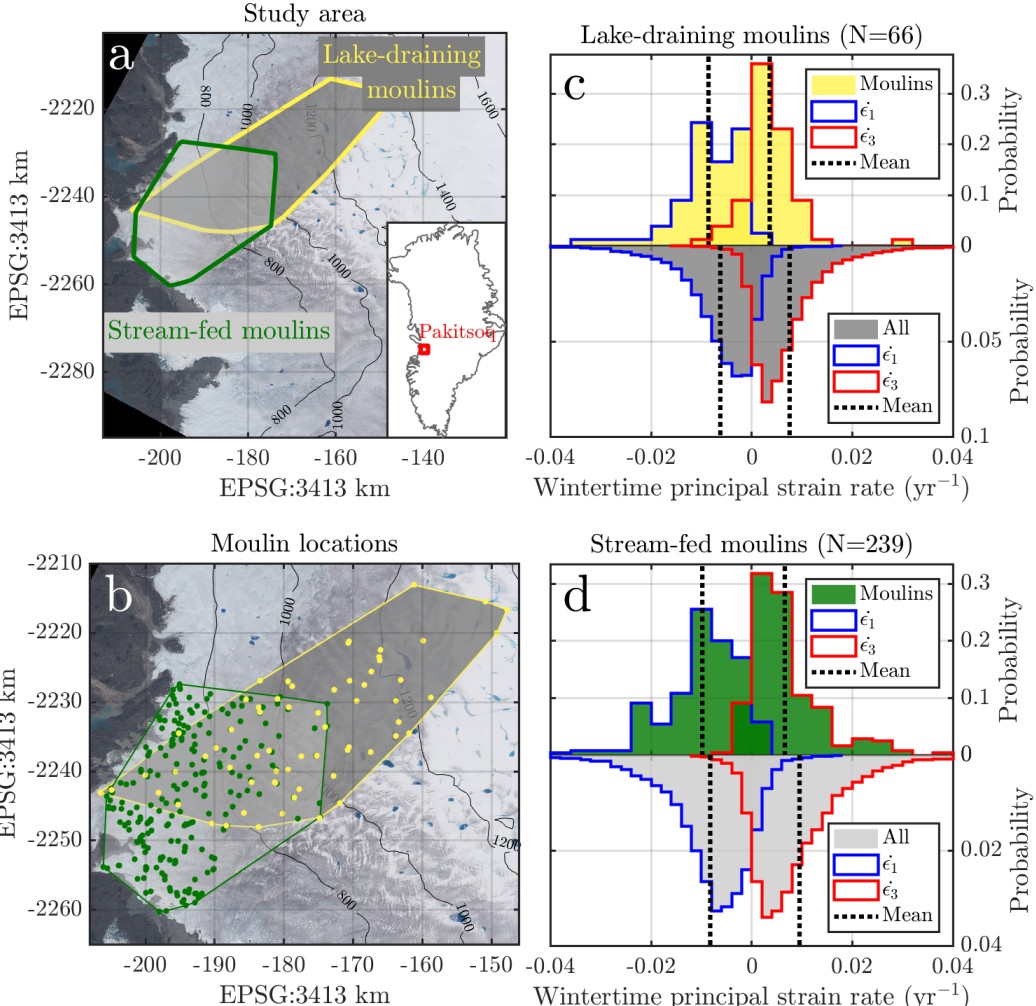

**Figure 3.** Study area and wintertime-average strain rates. (a) The Pâkitsoq area in western Greenland (inset), with the region where we have identified lake-draining moulins (Morriss et al., 2013) in gray and yellow, and the regions where we have identified stream-fed moulins (Andrews, 2015) in light gray and green. (b) Locations of the 78 lakes (yellow) from Morriss et al. (2013) and 239 stream-fed (green) moulins identified in 2011 (Andrews, 2015; Hoffman et al., 2018). We identified mean locations of the moulins that drain 66 of the 78 lakes over 2009–2019 (not shown). (c) Long-term average wintertime strain rates $\dot{\epsilon}_1$ and $\dot{\epsilon}_3$ at the 66 mean locations for lake-draining moulins (yellow), and at all locations in the lake-draining region (gray). (d) Long-term average wintertime strain rates at the 239 stream-fed moulins and at all locations in the stream-fed region (gray). Means are shown as black dashes and are significantly lower at moulin sites ($p < 0.05$) than the regional background, for both $\dot{\epsilon}_1$ and $\dot{\epsilon}_3$, and for lake-draining moulins and stream-fed moulins. Strain rates are calculated from the NASA MeASUREs Multi-year Greenland Ice Sheet Velocity Mosaic, Version 1 (Joughin et al., 2016a, b).





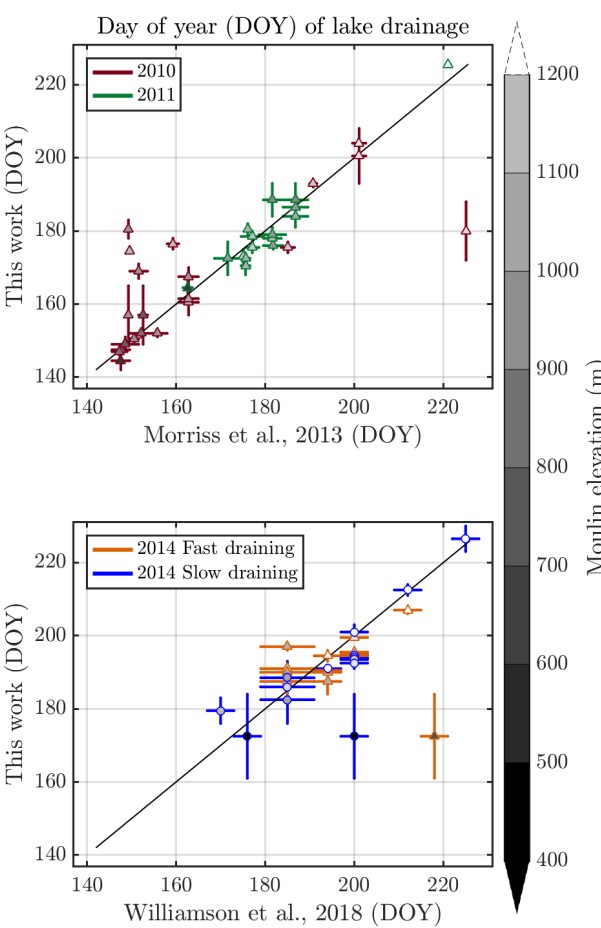

**Figure 4.** Comparison of lake drainage dates in this work to previous work, in the years that we have overlap. Elevations of each moulin Bamber:2013gw are shown by the gray tone, and marker shape indicates lake drainage type: triangles show fast-draining lakes, circles show slow-draining lakes. (a) Comparison to Morriss et al. (2013) over the years 2010–2011. Drainage dates of 22 out of 33 lakes (67%) agree to within uncertainties. (b) Comparison to Williamson et al. (2018b) over 2014. Drainage dates of 16 out of 21 (76%) lakes agree to within uncertainties.





**Figure 5.** Map of year-by-year moulin locations for a single lake for seven years between 2009–2017. (a) Regional setting of Lake #31, located at 69.585°N, 49.600°W in the northern part of the Pâkitsoq study area. Panel is 60×80 km. (b–h) Seven 700×1000-m WorldView images from 2009, 2010, 2012, and 2014–2017 showing the location of the lake-draining moulin in that year (colored dot) and in all other years (white dots). The moulin migrates over a ~350 m span over this nine-year period, reaching as far as ~700 m from the center of Lake #31. Lake drainage date, type, and completeness are noted on each panel. Ice flow is from right to left (east to west) at ~40 m/yr Joughin:2016jh, Joughin:2016tj. Scale bars show 100-m segments. (i) Histogram of lake – moulin distance for all $N = 95$ moulins with locations identified in WorldView images, 2009–2019, that were connected to fast-draining lakes.



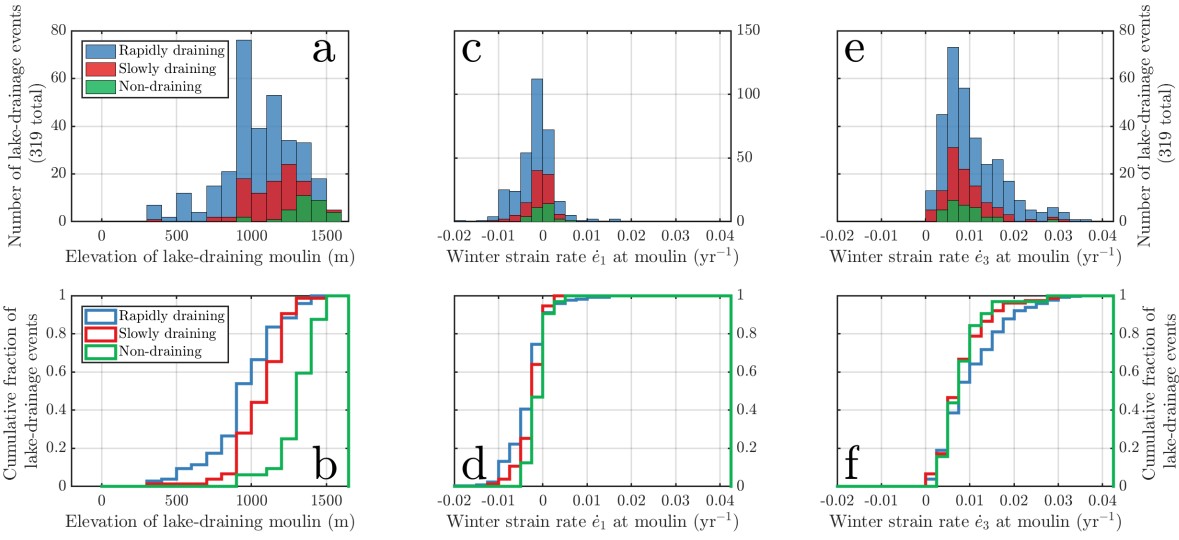

**Figure 6.** Distributions of elevation and strain rates at the locations of moulins associated with 78 supraglacial lakes, classified by drainage type of the lake (fast, slow, or non-draining). Moulins were located each year using WorldView imagery, when possible; when imagery was not available or when the lake did not drain, the moulin location was set to the center of the lake basin. (a) Histogram of elevations of the moulins at $N = 319$ lake drainage (or non-drainage) events over the 19 melt seasons from 2006–2010 and 2013–2018 whose type was observed with high confidence. (b) Cumulative distribution of the elevations of these moulins, separated by lake-drainage type. (c) Background more-compressive strain rate, $\dot{\epsilon}_1$, observed the preceding winter for each lake-draining moulin. (d) Cumulative distribution of $\dot{\epsilon}_1$, by lake-drainage type. (e) Background more-extensive strain rate, $\dot{\epsilon}_3$, observed the preceding winter for each lake-draining moulin. (f) Cumulative distribution of $\dot{\epsilon}_3$, by lake-drainage type. Strain rates are calculated with the near-yearly wintertime mosaics, MEaSUREs Greenland Ice Sheet Velocity Map from InSAR Data, Version 2 (Joughin et al., 2018b, 2010), supplemented with ESA Climate Change Initiative in winters 2013 and 2017 (Boncori et al., 2018) when MEaSUREs data are not available, as shown in Fig. 1.

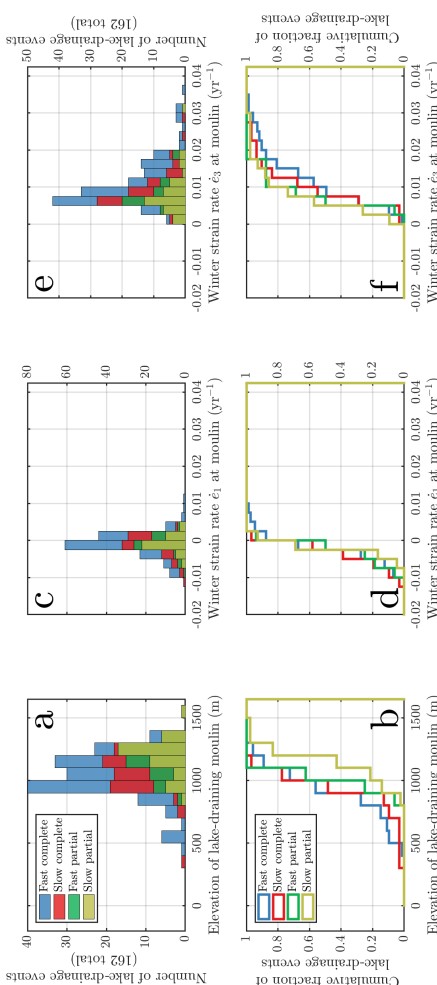

**Figure 7.** Distributions of elevation and strain rates at the locations of moulins associated with 78 draining supraglacial lakes, classified by drainage type of the lake (fast or slow) and the completeness of the lake drainage (complete or partial), following Chudley et al. (2019). Moulins were located each year using WorldView imagery, when possible; when imagery was not available or when the lake did not drain, the moulin location was set to the center of the lake basin. (a) Histogram of elevations of the moulins associated with $N = 162$ lake drainage events over the 19 melt seasons from 2006–2010 and 2013–2018 whose type and completeness were observed with high confidence. (b) Cumulative distribution of the elevations of these moulins, separated by lake-drainage type and completeness. (c) Background more-compressive strain rate, $\dot{\epsilon}_1$, observed the preceding winter for each lake-draining moulin. (d) Cumulative distribution of $\dot{\epsilon}_1$, by lake-drainage type and completeness. (e) Background more-extensive strain rate, $\dot{\epsilon}_3$, observed the preceding winter for each lake-draining moulin. (f) Cumulative distribution of $\dot{\epsilon}_3$, by lake-drainage type and completeness. Strain rates were calculated with the near-yearly wintertime mosaics, MEaSUREs Greenland Ice Sheet Velocity Map from InSAR Data, Version 2 (Joughin et al., 2018b, 2010), supplemented with ESA Climate Change Initiative in winters 2013 and 2017 (Boncori et al., 2018) when MEaSUREs data are not available, as shown in Fig. '.

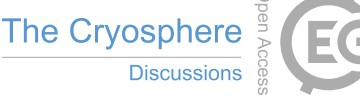

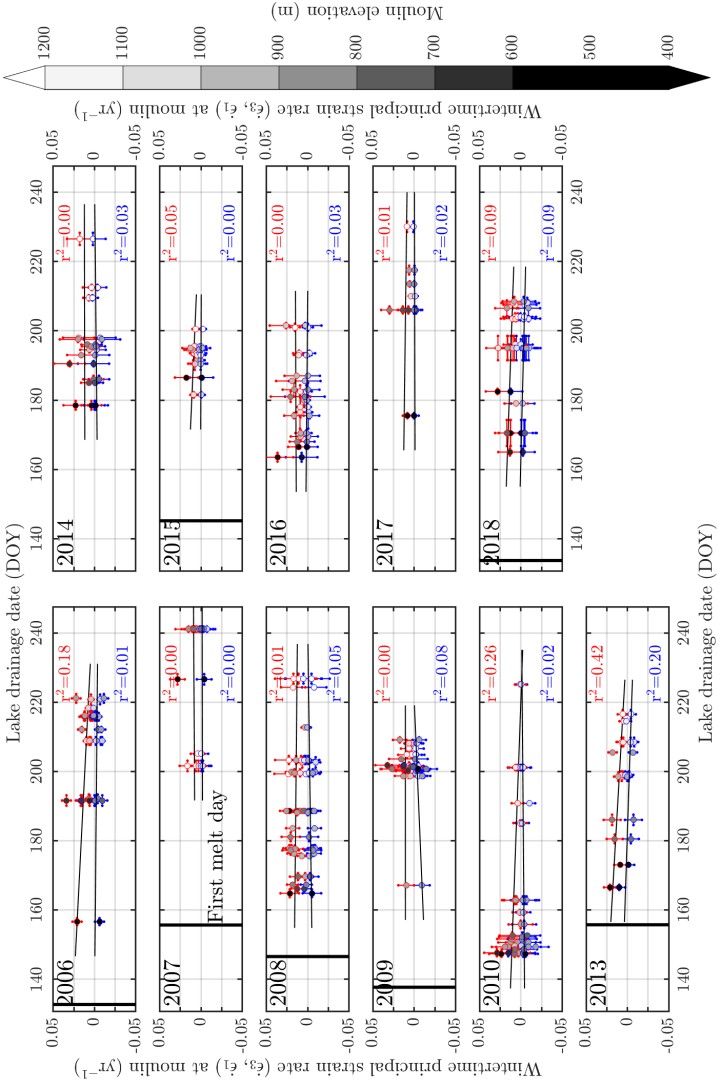

**Figure 8.** Wintertime strain rates at moulins associated with fast-draining lakes identified with high confidence, versus the dates of lake drainage. Strain rates are calculated from SAR-based velocity mosaics (Joughin et al., 2018b, 2010) for the winter immediately preceding the summer of lake drainage observations – i.e., the first panel shows strain rates at moulin sites observed in winter 2005–2006 versus the day of year 2006 that the fast-draining lake associated with each moulin drained (see Fig. 1). Blue points show $\dot{\epsilon}_1$, the more-compressional principal strain rate; red points show $\dot{\epsilon}_3$, the more-extensional principal strain rate. Thin black lines show the least-squares fits to each time series with the Pearson correlation coefficients, $r^2$, indicated. Bold black vertical lines show the day of each year that the 2-m air temperature first exceeded 0°C at the Swiss Camp meteorological station (Steffen et al., 1996); in some years (e.g., 2010), this occurs earlier than the range of the graph. Elevations of each moulin (Bamber et al., 2013) are shown by the gray tone.

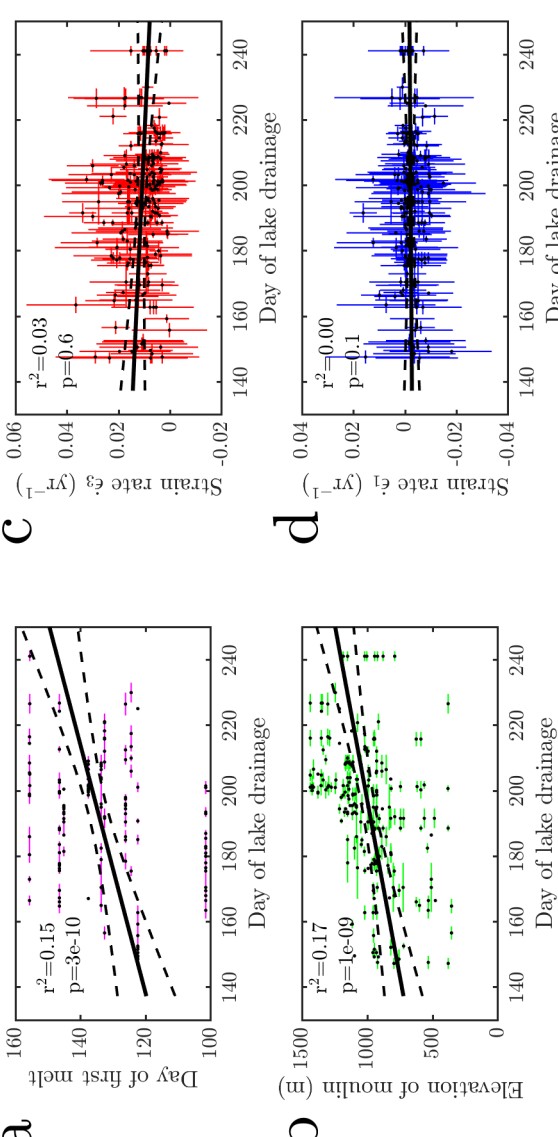

**Figure 9.** Results of four univariate regressions for lake drainage date for $N = 212$ fast lake drainage events identified with high confidence over the 19 melt seasons from 2006–2010 and 2013–2018. Thick black lines show the univariate least-squares fits to each time series, dashed lines show the 99% confidence bands on each fit, and the Pearson correlation coefficients ($r^2$) and $p$-values from a separate multivariate regression are indicated. (a) Date of melt onset versus the day of fast lake drainage for these 11 years (dots), with uncertainties on the date of lake drainage (magenta). (b) Scatter plot of moulin elevation versus the day of drainage of the fast-draining lake each moulin is connected to (dots), with uncertainties on the date of lake drainage (green). (c) Scatter plot of background extensional principal strain rate $\dot{\epsilon}_3$ versus the day of drainage of the fast-draining lake each moulin is connected to (dots), with uncertainties on the strain rate and on the date of lake drainage (red). (d) Scatter plot of background compressional principal strain rate $\dot{\epsilon}_1$ versus the day of drainage of the fast-draining lake each moulin is connected to (dots), with uncertainties on the strain rate and on the date of lake drainage (blue).



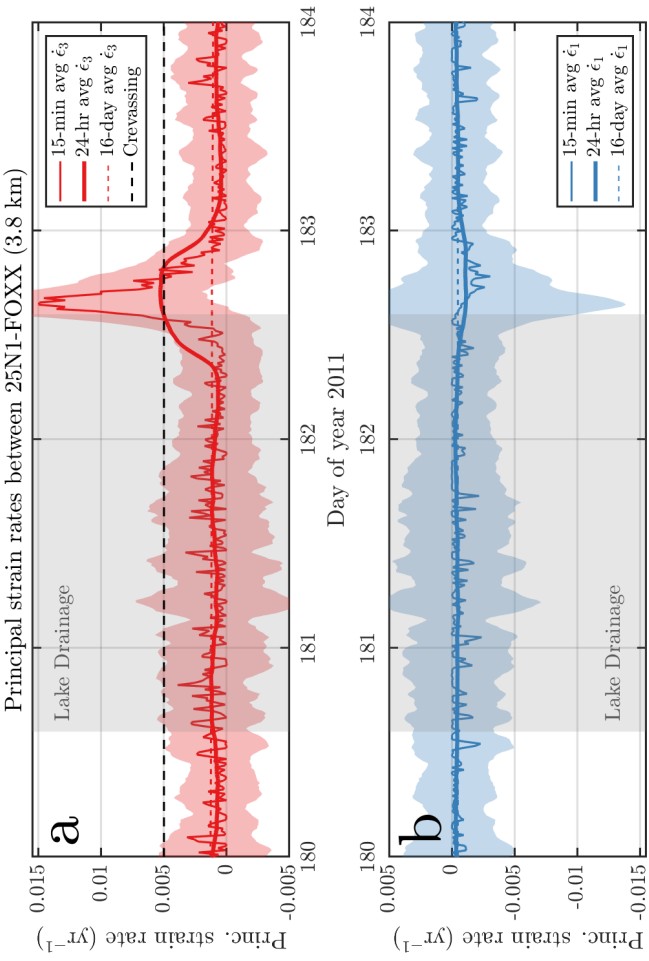

**Figure 10.** GPS-derived principal strain rates, $\dot{\epsilon}_1$ (bottom panel) and $\dot{\epsilon}_3$ (top panel), averaged across a 3.8-km distance along an approximate basal hydropotential flowpath over four days of summer 2011. Three nearby fast lake drainages (Lakes #52, 55, 64) occurred on Day $181.6 \pm 1$ days, according to Morriss et al. (2013) and this work. These lakes are respectively 12 km, 10 km, and 8 km upstream from the observation midpoint between GPS stations shown here. The shaded region shows the estimated error in strain rates (Equation 3), based on GPS location errors of approximately 0.5 cm for each observation recorded at 15-minute intervals. The dashed black line shows the approximate strain-rate threshold for crevassing, $+0.005 \text{ yr}^{-1}$ (Poinar et al., 2015; Joughin et al., 2013). GPS data are shown at 15-minute, 24-hour, and 16-day averaging intervals. GPS data are from Andrews et al. (2018) and Hoffman et al. (2018).





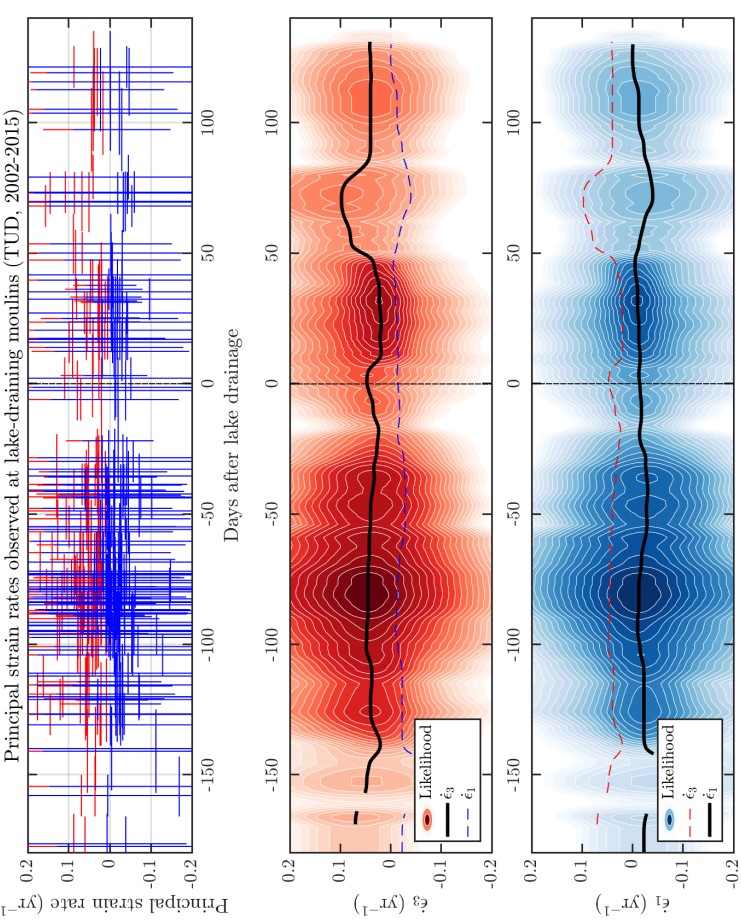

**Figure 11.** (a) Strain rates at moulins connected to 18 fast-draining lake events over the period 2002–2015 in the Pâkitsoq region below ~900 m elevation, calculated from the Landsat-derived velocity product from the Technical University at Dresden (Rosenau et al., 2015) with time resolution of 16–32 days. Blue lines are more-compressional principal strain rates, with error bars in strain rate and time; red lines are more-extensional principal strain rates. (b) The number of observed more-extensional principal strain rates, $\dot{\epsilon}_3$, displayed as a heat map, with number of observations (colored background) versus strain rate magnitude (y-axis) and number of days before or after each individual fast-draining lake event (x-axis). (c) The number of observed more-compressional principal strain rates, $\dot{\epsilon}_1$, displayed as a heat map. These data represent 561 velocity scene pairs, differenced to obtain regional strain-rate observations and interpolated onto 18 moulin sites before, during, and after 97 fast-draining lake events into those moulins over this 14-year period. The black, red, and blue lines plot the time series of the most likely principal strain rates $\dot{\epsilon}_1$ and $\dot{\epsilon}_3$, inferred directly from the data shown in the heat maps.



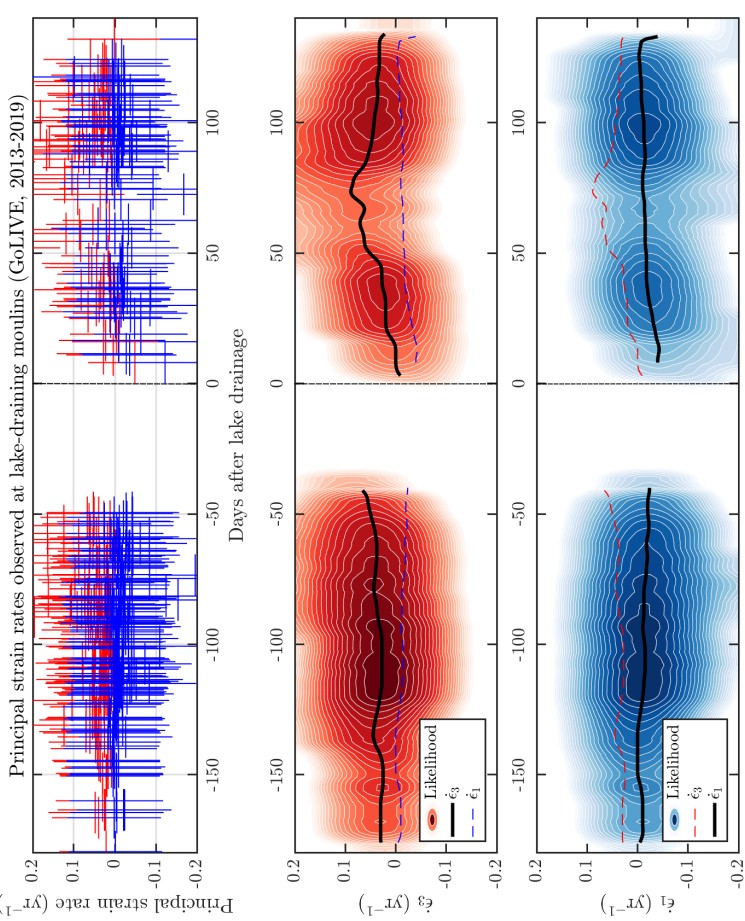

**Figure 12.** (a) Strain rates at moulins connected to 144 fast-draining lake events over the period 2013–2019, calculated from the GoLIVE Landsat-8-derived velocity product (Scambos et al., 2016; Fahnestock et al., 2016) with time resolution of 16 days. This dataset covers the entire Pâkitsoq region. (b) The number of observed more-extensive principal strain rates, $\dot{\epsilon}_3$, displayed as a heat map, with number of observations (colored background) versus strain rate magnitude (y-axis) and number of days before or after each individual fast-draining lake event (x-axis). (c) The number of observed more-compressive principal strain rates, $\dot{\epsilon}_1$, displayed as a heat map. These data represent 97 velocity scene pairs, differenced to obtain regional strain-rate observations and interpolated onto the moulin sites before, during, and after 144 fast-draining lake events into those moulins over this 7-year period. The black, red, and blue lines plot the time series of the most likely principal strain rates $\dot{\epsilon}_1$ and $\dot{\epsilon}_3$, inferred directly from the data shown in the heat maps. The presence of water during the melt season interferes with the production of the GoLIVE observations from roughly 40 days before most lake drainages through roughly 10 days after.



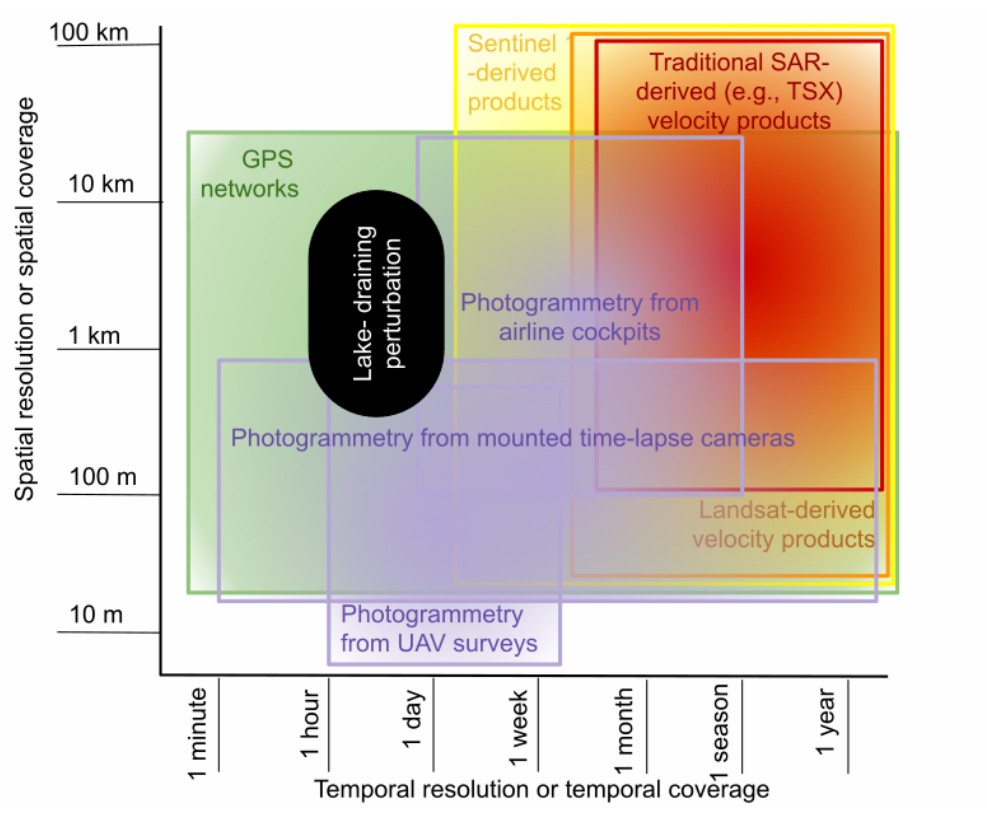

**Figure 13.** Illustration of the rough spatial and temporal resolutions of candidate methods and datasets (colors) evaluated here for resolving hypothesized lake-drainage precursor events (black) at the regional scale.





**Table 1.** Candidate methods for transient strain rate detection, along with their characteristics in spatial coverage, spatial resolution, temporal coverage, and temporal resolution. Checks and strike marks in the last column indicate whether the method is sufficient, in each respective category, to resolve hypothesized lake-drainage precursor events at the regional scale. Question marks denote open avenues for progress.

| Candidate sensor | Spatial coverage | Spatial resolution | Temporal coverage | Temporal resolution | Overall suitability |
|---|---|---|---|---|---|
| Single GPS station | point | point | ~6 months/yr | 15 seconds | ✗ ✗ ✓ ✓ |
| GPS network: Andrews et al. (2018) | ~20 km | > 2 km | ~2 melt seasons | 15 seconds | ✗ ✗ ✓ ✓ |
| GPS network: Stevens et al. (2015) | ~10 km | ~1 km | 3 melt seasons | 30 seconds | ✗ ✓ ✓ ✓ |
| MEaSUREs SAR-derived velocity products | > 100 km | 100–250 m | > 10 years | 11 days – 1 year | ✓ ✓ ✓ ✗ |
| Landsat-derived velocity products | > 100 km | 150–300 m | Since 2013 (L8) | ≥ 16 days | ✓ ✓ ✓ ✗ |
|  |  |  | Since 1985 (L7) |  | ✓ ✓ ✓ ✗ |
| Sentinel 1a/b-derived velocity products | > 100 km | ≥ 300 m | Since 2014 | ≥ 6 days | ✓ ✓ ✓ ✗ |
| Sentinel 1a/b/c/d-derived velocity products |  |  | Planned launch 2022 | ≥ 1.5 days | ✓ ✓ ✓ ✗ |
| Airborne photogrammetry | 10–30 km | < 1 km | Seasonal | Near-daily | ? ✓ ✓ ? |
| Regional GPS network |  |  | ~6 months/yr | 30 seconds | ? ? ✓ ✓ |
| Ideal sensor for regional-scale lake-drainage-inducing strain rate perturbations | > 30 km | < 1 km | >2 months/yr | ≤ 24 hours | ✓ ✓ ✓ ✓ |