# Peer review of "Challenges in predicting Greenland supraglacial lake drainages at the regional scale"

_The Cryosphere, 2020_

## Referee Comment (RC1) · Anonymous Referee #1 · 12 Nov 2020

**Summary**

Poinar and Andrews present a new analysis exploring the hypothesised links between supraglacial lake drainages on the Greenland Ice Sheet and the influence of both background and transient stresses. Using remotely sensed lake drainage histories and strain rate fields derived from publicly available velocity products, they find that fast-draining lakes are associated with significantly more-extensional background strain rates than slow-draining or non-draining lakes, although this relationship does not extend to the date of drainage. They show that 16-32 day remotely sensed velocity observations are not useful for identifying hypothesised transient stresses, and make

several alternative recommendations as to how data on such events may be collected in the future, and ultimately implemented into ice sheet models.

I believe this paper is a unique and important contribution as it goes some way to addressing questions raised by recent work on supraglacial lakes on the Greenland Ice Sheet, synthesising issues raised by field-based, remote sensing, and modelling studies. The manuscript is well written and logically structured. Furthermore, the authors do an excellent job of explaining the methods and background data, and I see this paper being additionally useful as general reference for those wishing to take advantage of the recent explosion of publicly available Greenland velocity data.

**Specific Comments**

The authors equate the two surface-parallel principal strains to the maximum and minimum principal strains ($\epsilon_1$ and $\epsilon_3$), assuming that the principal strain normal to the surface (with a value of 0 yr$^{-1}$) is always intermediate between the two surface-parallel values (and thus is always $\epsilon_2$). However, Vaughan (1993) identifies that on an ice surface with open fractures (which is thus not incompressible) there are situations where observations can show surface-parallel principal strains to be both positive or both negative. As such, the zero normal stress may be any one of the maximum, intermediate, or minimum principal stresses. Consider instead explicitly defining the surface-parallel components as simply $\epsilon_1$ and $\epsilon_2$ (or, is more precision is desired, $\epsilon_{1surf}$ and $\epsilon_{2surf}$), disregarding the vertical component (see also Hooke, 1998 or Doake et al. 1998 for examples of this).

The authors separate lakes into completely and partially draining types (L200-205) following Chudley et al. (2019). However, Chudley et al. make no explicit recommendation as to parameters that may separate these lake types, and as such the 10% threshold has been chosen by the authors. Given the established sensitivity of lake drainage studies to chosen parameters (Cooley and Christoffersen, 2017), it might be desirable to include, perhaps as a supplement, data showing the sensitivity of the classification to varying this threshold by some percent.

I have some queries regarding Section 4.1.2, in particular the statement 'fast-draining... and bottom-draining are not synonyms' (L514). Probably originating from the binary described by Tedesco et al. (2013), I have always considered 'fast-draining' and 'bottom-draining' to be synonymous (i.e. to indicate a lake that has drained in a matter of hours following hydrofracture of the lake-bed), as well as 'slow-draining' and 'overtopping' (i.e. a lake that has drained in a matter of days following progressive incision of an outlet channel). Indeed this synonymy is made explicit in definitions included by e.g. Banwell et al. (2012), Selmes et al. (2013), Fitzpatrick et al. (2014), Koziol et al. (2017), and Williamson et al. (2018). My reading of nearly all remote sensing studies is that any 'fast-draining' threshold (e.g. <6 days for this study) is simply the best available method of trying to differentiate the underlying physical mechanisms (hydrofracture vs. overtopping). If I were to observe that '40% of the lakes we classify as fast-draining are not bottom-draining' (L494), I would see that as evidence of classification error (e.g. the lakes drained slowly via overtopping but in 4 days, so were missed by the 6-day threshold) rather than evidence that the two terms are not synonymous. The only situation I can imagine to the contrary would be a situation where an overtopping lake induced non-local hydrofracture and drained in a matter of hours - however, I cannot see how the data presented in this study supports such an inference, as 'fast-draining' is defined using only the 6-day threshold. Of course, this debate could be seen as rather academic, as whether people are using these as synonyms does not change the underlying processes - however, considering the importance of these definitions to both methods and mechanisms, most of all perhaps this is evidence that as a community we should be making more effort to ensure we're all on the same page with regards to these terms.

One final thing that I do not believe is commented on is the interannual variation of individual lakes. Are most lakes in the dataset draining in uniform ways every year (e.g. always rapid, always non-draining) or is it more variable? Can this also be related

to background strain?

**Minor Comments**

- L30 - Cite also Doyle et al. 2013 here.

- Paragraph beginning L35 - Mention also Hoffman et al. 2018 here.

- L65-66 - Cite also Sugiyama et al. 2008 here.

- L146 - "These definitions follow Harper and Humphery (Harper et al. 1998)". Surely just "...follow Harper et al. (1998)" or "...Harper and Humphrey ([year])"?

- L206-210 - The methods are largely excellent, but more information should be included as the classification procedure for high, moderate, and low confidence levels, which are irreproducible from this text alone.

- L424 - Can this increase be shown to be statistically significant? I find it hard to believe that it can, especially considering the paucity of data in the days preceding.

- L491-492 - The authors identify bottom-draining moulins as being within 390 m of the lake center, justified as being the average radius of the sample lakes. Whilst I understand that identifying bottom-draining moulins for individual lakes from their respective extents may be too much work, it would be useful to include the standard deviation radius or some other measure of variance, so that the reader can judge the extent to which using the average is helpful.

- L591, and elsewhere - Some errors with bibtex or equivalent citation software are occurring here.

- L636 - 0.3 km$^2$ seems a bit small for an entire lakebed study?
- L637-639 - Arguably the spatial coverage here is slightly too limiting - Jouvet et al. (2019) have shown that the typical UAVs used in the Greenlandic literature can be effectively upscaled to an endurance of 3 hours / 180 km, able to cover one large study site, or multiple different study sites, at a distance from the operator. In this context, I would argue that the spatial coverage of UAVs in Fig. 13 can be upped to 10 km. This is without considering high altitude, long endurance (HALE) UAVs that effectively blur the line between UAV and aircraft, although of course these are largely beyond the engineering and logistical competencies of an individual glaciological research group. For a convincing application, however, see Crocker et al. (2011), who were able to make glaciological observations over three lakes 100 km away from the comfort of Ilulissat.

- L663 - Perhaps considering whether the recent abundance of low-cost carrier-phase GNSS, as well as recent advances such as the L2C band, make a comprehensive low-cost network more feasible for the next decade than previously.

- Paragraph beginning L674 - This review of surface routing models misses that of Koziol et al. (2017).

Fig 5: It would be useful to add colours to each moulin point to indicate the year of drainage, as well as an arrow indicating flow direction to each panel. This would make it easier to identify recurring moulins as discussed in Section 3.2 and elsewhere.

**References not in the main text**

Crocker, R. I., Maslanik, J. A., Adler, J. J., Palo, S. E., Herzfeld, U. C., Emery, W. J. (2011). A sensor package for ice surface observations using small unmanned aircraft systems. IEEE transactions on geoscience and remote sensing, 50(4), 1033-1047.

Doake, C. S. M., Corr, H. F. J., Rott, H., Skvarca, P., Young, N. W. (1998). Breakup and conditions for stability of the northern Larsen Ice Shelf, Antarctica. Nature, 391(6669), 778-780.

Hooke, R. L. (1998). Principles of glacier mechanics. Prentice Hall.

Jouvet, G., Weidmann, Y., van Dongen, E., Luethi, M., Vieli, A., Ryan, J. (2019). High-endurance UAV for monitoring calving glaciers: Application to the Inglefield Bredning and Eqip Sermia, Greenland. Frontiers in Earth Science, 7, 206.

Sugiyama, S., Bauder, A., Huss, M., Riesen, P., Funk, M. (2008). Triggering and drainage mechanisms of the 2004 glacier‐dammed lake outburst in Gorner-gletscher, Switzerland. Journal of Geophysical Research: Earth Surface, 113(F4).

Vaughan, D. G. (1993). Relating the occurrence of crevasses to surface strain rates. Journal of Glaciology, 39(132), 255-266.

---

## Referee Comment (RC2) · Anonymous Referee #2 · 13 Nov 2020

Supraglacial lake drainage has crucial impacts on surface-to-bed meltwater connection on the Greenland Ice Sheet but remains challenging to quantify. This study uses remote-sensing velocity datasets to constrain the relationship between strain rates and supraglacial lake drainage and to test the hypothesis that transient strain rates drive fast lake drainage. The results show significantly more-extensional background strain rates at moulins associated with fast-draining lakes than at slow-draining or non-draining lake moulins. This study aims to solve an important science question for the Greenland research community. I recommend it for publication with some minor changes.

General comments:

[Figure]

(1) The structure of the paper is basically clear but can be improved. Some suggestions: Section 3.4, this section is short so it may not be necessary to divide it into two sub-sections. Section 3.6, some descriptions belong to methods and should be removed. Section 3.7, the first paragraph of this section belongs to methods and should be removed. The discussion section presents very insightful ideas but the discussion should be based on the results of this study. I think sections "4.2 Prediction of future lake-drainage events" and "4.3 Parameterizing moulins in ice-sheet models" should be more closely related to the main findings of this study. In other words, these two sections should highlight how the findings of this study can help us better answer the two crucial science questions (lake drainage and new generation of ice sheet models) rather than broadly introducing these two science questions. This can be done by slightly modifying some words and expressions.

(2) Significance test is widely used in the study. It may be useful to briefly explain how the test was conducted at different parts of the results.

(3) The paper concludes that "observational progress in understanding lake drainage initiation will rely on field-based tools such as GPS networks and photogrammetry". I think this should be further discussed. A growing availability of high-resolution satellite imagery (e.g. CubeSat and Landsat-9) provide more frequent observations of supraglacial lakes in future and may mitigate the time gap problem.

(4) The study area of this paper is relatively small ($\sim$1600 km2) and most cover low elevations (<1400 m). Will the results obtained in this study be applicable for larger areas? particularly when including high-elevation areas. It will be useful to briefly discuss this point.

Specific comments:

line 16, Smith et al (2015) found nearly all surface meltwater drain into moulins in the ablation zone of the southwestern GrIS rather than in the western GrIS. I think it is necessary to distinguish these two study areas. It is not clear if all meltwater drains

to moulins, particularly at the high-elevation areas of the western GrIS since few river, lake, or moulin maps have been made for this region.

line 18, on diurnal scale too.

line 20, add "." before "Our".

line 28, "basins of specific supraglacial lakes", do "basins" mean the topographic depressions that host lakes or the upstream contributing catchment area to feed lakes?

line 42, supraglacial river gauging, streams are narrow and exhibit small contributing areas.

line 50, Banwell and Sommers are not appropriate to describe "the next generation of ice sheet models".

lines 59-61, this sentence is not easy to follow.

lines 102-103, how to obtain velocity uncertainties?

line 121, panchromatic pixels?

line 232, panchromatic band? Multi-spectral bands have lower spatial resolution (2 m).

line 258, how is p value calculated?

line 266, e3?

line 285, are most of these moulins located in topographic depressions that host lakes?

lines 308-310, how about comparing to Cooley and Christoffersen (2017)?

lines 320-322, how is p value calculated?

line 421, standard deviation 15 days is a very long time because most supraglacial lakes have short lifetime spans. Any implications we can obtain from this long std?

line 428, what does "meaningfully change" mean?

line 591, fix "*Stevens:2015ht".

line 723, "meltwater" rather than "melt" delivery to the bed.

Figure 4, the moulin elevation colors are not clear for the dots, perhaps change into color ramp? Fix "Bamber:2013 gw" in the figure caption.

Figure 8, see the comment for Figure 4.
* * *

---

## Author Comment (AC1) · 30 Dec 2020

*Authors' replies inline in red    December 29, 2020*

**Summary**

Poinar and Andrews present a new analysis exploring the hypothesised links between supraglacial lake drainages on the Greenland Ice Sheet and the influence of both back-ground and transient stresses. Using remotely sensed lake drainage histories and strain rate fields derived from publicly available velocity products, they find that fast-draining lakes are associated with significantly more-extensional background strain rates than slow-draining or non-draining lakes, although this relationship does not ex-tend to the date of drainage. They show that 16-32 day remotely sensed velocity observations are not useful for identifying hypothesised transient stresses, and make several alternative recommendations as to how data on such events may be collected in the future, and ultimately implemented into ice sheet models. I believe this paper is a unique and important contribution as it goes some way to addressing questions raised by recent work on supraglacial lakes on the Greenland Ice Sheet, synthesising issues raised by field-based, remote sensing, and modelling studies. The manuscript is well written and logically structured. Furthermore, the authors do an excellent job of explaining the methods and background data, and I see this paper being additionally useful as general reference for those wishing to take advantage of the recent explosion of publicly available Greenland velocity data.

**Specific Comments**

The authors equate the two surface-parallel principal strains to the maximum and minimum principal strains ($\dot{e}_1$ and $\dot{e}_3$), assuming that the principal strain normal to the surface (with a value of 0 yr$^{-1}$) is always intermediate between the two surface-parallel values (and thus is always $\dot{e}_2$). However, Vaughan (1993) identifies that on an ice surface with open fractures (which is thus not incompressible) there are situations where observations can show surface-parallel principal strains to be both positive or both negative. As such, the zero normal stress may be any one of the maximum, intermediate, or minimum principal stresses. Consider instead explicitly defining the surface-parallel components as simply $\dot{e}_1$ and $\dot{e}_2$ (or, is more precision is desired, $\dot{e}_{1surf}$ and $\dot{e}_{2surf}$), disregarding the vertical component (see also Hooke, 1998 or Doake et al. 1998 for examples of this).

*This idea improves the communication of our strain rates. We have implemented it and explained the reasoning (lines ~150 in the differenced document). We have also updated Figures 3, 6, 7, 8, 9, 10, 11, and 12 , as well as all appearances of $\dot{e}_3$ in the text, to incorporate the improved nomenclature.*

The authors separate lakes into completely and partially draining types (L200-205) following Chudley et al. (2019). However, Chudley et al. make no explicit recommendation as to parameters that may separate these lake types, and as such the 10% threshold has been chosen by the authors. Given the established sensitivity of lake drainage studies to chosen parameters (Cooley and Christoffersen, 2017), it might be desirable to include, perhaps as a supplement, data showing the sensitivity of the classification to varying this threshold by some percent.

*Yes, the 10% threshold is our choosing. We assigned it based on the figures in Chudley et al. (2019) and visual inspection of the Landsat images on which (in part) we based our lake-drainage dataset. Exploring the sensitivity of our results to different thresholds, such as perhaps 50%, should be possible by revisiting each of our high-confidence fast- or slow-draining lakes (N=287) and reclassifying any partial drainages, which we identify by estimating the lake area change across consecutive Landsat images by eye. We do not think this sensitivity testing would significantly change our results or interpretation, which center on distinguishing fast lake drainages from slow drainages or non-draining lakes, rather than their completeness. The treatment of these topics in our Results and Discussion sections – lake drainage speed (~5 subsections) and lake drainage completeness (~2 subsections) – underscores this relative emphasis. Nonetheless, the suggestion for analysis of complete/partial sensitivity is potentially meaningful and is something we will consider for future work.*

I have some queries regarding Section 4.1.2, in particular the statement 'fast-draining...and bottom-draining are not synonyms' (L514). Probably originating from the binary described by Tedesco et al. (2013), I have always considered 'fast-draining' and 'bottom-draining' to be synonymous (i.e. to indicate a lake that has drained in a matter of hours following hydrofracture of the lake-bed), as well as 'slow-draining' and 'overtopping' (i.e. a lake that has drained in a matter of days following progressive incision of an outlet channel). Indeed this synonymy is made explicit in definitions included by e.g. Banwell et al. (2012), Selmes et al. (2013), Fitzpatrick et al. (2014), Koziol et al. (2017), and Williamson et al. (2018). My reading of nearly all remote sensing studies is that any 'fast-draining' threshold (e.g. <6 days for this study) is simply the best available method of trying to differentiate the underlying physical mechanisms (hydrofracture vs. over-topping). If I were to observe that '40% of the lakes we classify as fast-draining are not bottom-draining' (L494), I would see that as evidence of classification error (e.g. the lakes drained slowly via overtopping but in 4 days, so were missed by the 6-day threshold) rather than evidence that the two terms are not synonymous. The only situation I can imagine to the contrary would be a situation where an overtopping lake induced non-local hydrofracture and drained in a matter of hours - however, I cannot see how the data presented in this study supports such an inference, as 'fast-draining' is defined using only the 6-day threshold. Of course, this debate could be seen as rather academic, as whether people are using these as synonyms does not change the underlying processes - however, considering the importance of these definitions to both methods and mechanisms, most of all perhaps this is evidence that as a community we should be making more effort to ensure we're all on the same page with regards to these terms.

*We agree with these insightful comments. The Chudley et al. (2019) study blew my (K.P.) mind as well and similarly made me reframe my conception of "fast-draining" and "bottom-draining" lakes, with the new dimension of "complete" versus "partial" drainage. I agree with your assessment that earlier studies (Banwell et al., 2012 through Williamson et al., 2018, including perennially influential ones such as Tedesco et al., 2013) used drainage speed as a proxy for drainage mechanism. With the new wide*

*availability of WorldView and frequent Sentinel-2 imagery, I think we can soon move beyond this, at least for smaller studies such as this one (N=78 lakes) or perhaps with future AI approaches!*

*I'd like to address your point that "40% of the lakes we classify as fast-draining are not bottom-draining" (L494) could amount to classification error. I agree with your suggestion that an overtopping lake feeding a non-lake-bottom hydrofracture and draining in something like ~4 days (or possibly even a matter of hours) would satisfy this classification scenario. I think the point of disparity is "non-lake-bottom" versus "non-local" hydrofracture. Field observations show that hydrofractures some ~500 m (Chudley et al., 2019) or ~1500 m (Stevens et al., 2015) from the lake bottom can facilitate fast lake drainages. In both studies, these hydrofractures sat within the lake basin, but closer to the downstream edge. In both cases, our simple analysis would categorize these as "non-bottom-draining" events because the hydrofractures are both >390 meters from the lake center. Thus, our classification has two problems: (1) using the mean lake radius of 390 meters as a threshold, and (2) approximating the lake bottom as the lake center. I propose to redefine "non-bottom-draining" as feeding a moulin more than mean + 2sigma lake radius (700 meters) from the lake center, in an effort to account for lake-to-lake variability and uncertainty in the precise location of the lake bottom. This changes the sentence in question to "some 10–20% of the lakes we classify as fast-draining are not bottom-draining". We've updated the text in Section 4.1.2 accordingly (lines ~545 in the differenced document).*

One final thing that I do not believe is commented on is the interannual variation of individual lakes. Are most lakes in the dataset draining in uniform ways every year (e.g. always rapid, always non-draining) or is it more variable? Can this also be related to background strain?

[Figure]

*See the figure here, which summarizes the drainage type by lake and by year of our entire dataset. There is substantial year-to-year variability in drainage type at many of the 78 lakes. In general, higher-elevation lakes have lower indices (~#1–20, toward the top of the diagram), and lower-elevation lakes have higher indices (~#60–78, toward the bottom), but the indices are not carefully ordered (Morris et al., 2013). You can see that higher-elevation lakes may undergo fast, slow, or non-drainage from year to year, while lower-elevation lakes are less variable. Some lakes could be called "usually fast" (e.g., Lake #49) or "usually slow" (e.g., Lake #14). Considering the already-large size of this paper and the limited insight*

*gleaned from this variability analysis, we're opting to limit its inclusion to only this response document.*

**Minor Comments**

L30 – Cite also Doyle et al. 2013 here.
>  *Added (line 32 in the differenced document).*

Paragraph beginning L35 – Mention also Hoffman et al. 2018 here.
>  *Added (line 37 in the differenced document).*

L65-66 – Cite also Sugiyama et al. 2008 here.
>  *Added (line 67 in the differenced document).*

L146 – "These definitions follow Harper and Humphery (Harper et al. 1998)". Surely just "...follow Harper et al. (1998)" or "...Harper and Humphrey ([year])"?
>  *Fixed (line 150 in the differenced document).*

L206-210 – The methods are largely excellent, but more information should be included as the classification procedure for high, moderate, and low confidence levels, which are irreproducible from this text alone.
>  *Explanation added (lines ~247 in the differenced document).*

L424 – Can this increase be shown to be statistically significant? I find it hard to believe that it can, especially considering the paucity of data in the days preceding.
>  *Indeed it is not significant. We added the words "but insignificant" to specify this (line 478 in the differenced document).*

L491-492 – The authors identify bottom-draining moulins as being within 390 m of the lake center, justified as being the average radius of the sample lakes. Whilst I understand that identifying bottom-draining moulins for individual lakes from their respective extents may be too much work, it would be useful to include the standard deviation radius or some other measure of variance, so that the reader can judge the extent to which using the average is helpful.
>  *See our long response to the earlier "specific comment" on lake-bottom moulins.*

L591, and elsewhere - Some errors with bibtex or equivalent citation software are occurring here.
>  *Fixed (line 650 in the differenced document).*

L636 – 0.3 km$^2$ seems a bit small for an entire lakebed study?
>  *Indeed. We read this incorrectly from the Methods of that study, which describes a planned image footprint of 400 x 660 meters = 0.29 km$^2$. From Figure 4 of Chudley et al. (2019), however, the UAV-mapped area looks more like ~40 km x 1.5 km = 60 km$^2$. We've accordingly replaced 0.3 km$^2$ with 60 km$^2$ (line 698 in the differenced document).*

L637-639 – Arguably the spatial coverage here is slightly too limiting - Jouvet et al. (2019) have shown that the typical UAVs used in the Greenlandic literature can be effectively upscaled to an endurance of 3 hours / 180 km, able to cover one large study site, or multiple different study sites, at a distance from

the operator. In this context, I would argue that the spatial coverage of UAVs in Fig. 13 can be upped to 10 km. This is without considering high altitude, long endurance (HALE) UAVs that effectively blur the line between UAV and aircraft, although of course these are largely beyond the engineering and logistical competencies of an individual glaciological research group. For a convincing application, however, see Crocker et al. (2011), who were able to make glaciological observations over three lakes 100 km away from the comfort of Ilulissat.

*We appreciate these UAV references. After reviewing them, we agree that a spatial coverage estimate of some tens of kilometers is more appropriate, which is in fact what we have in Table 1 ("10–30 km"), but we've now increased it to 10 km in Figure 13. We've also incorporated context from both studies into Section 4.2.3.2, Photogrammetry Observations, added these citations, and adjusted our assessment of the potential of airborne photogrammetry accordingly (lines ~701 in the differenced document).*

L663 – Perhaps considering whether the recent abundance of low-cost carrier-phase GNSS, as well as recent advances such as the L2C band, make a comprehensive low-cost network more feasible for the next decade than previously.

*We've added a discussion of these technologies to Section 4.2.3.C, "Dense regional GPS networks" (line ~725 in the differenced document), with the following text:*

*Our ability to accurately measure GPS receiver position and velocity on ice sheets has improved with the advent of carrier-phase technology, now used widely in glaciology (e.g., Ryser et al., 2014; Andrews et al., 2018; Jouvet et al., 2019; Riverman et al., 2019), and the 2013 implementation of the L2C band, which comes at the cost of power requirements to monitor both L1 and L2 bands (e.g., Van de Wal et al., 2015). Use of single-phase receivers can reduce instrument costs, power requirements, and instrument attrition, allowing deployment of more extensive or denser arrays (e.g., Van de Wal et al., 2015; Sutherland et al., 2015). However, these benefits must be balanced with reduced accuracy, which becomes critical for observing ice motion at hourly timescales, and increased maintenance needs. Design of any GPS network will require careful consideration of the trade-offs in spatial resolution, spatial coverage, and the cost and feasibility to install and maintain stations in the challenging conditions of ice-sheet ablation zones.*

Paragraph beginning L674 – This review of surface routing models misses that of Koziol et al. (2017).

*We meant this to be a summary of the input methods used in subglacial models, rather than a review of surface routing models. We see that the Clason et al. references blur that line, as that work is really an englacial, not subglacial, hydrology model. We now specify that the Clason et al. studies are englacial hydrology models (line 760 in the differenced document). We also added the Koziol et al. reference alongside Banwell et al. (2016), who uses essentially the same surface routing approach (line 755 in the differenced document).*

Fig 5: It would be useful to add colours to each moulin point to indicate the year of drainage, as well as an arrow indicating flow direction to each panel. This would make it easier to identify recurring moulins as discussed in Section 3.2 and elsewhere.

*We have added year labels to each moulin in every panel. We retained the white dots for "off-year" moulins to emphasize the "current-year" moulin, whose dot is colored. Overall, we think the change addresses the stated goal of making it easier to identify recurring moulins from panel to panel. We have also added a yellow ice-flow arrow to each panel.*

---

## Author Comment (AC2) · 30 Dec 2020

*Authors' replies inline in red     December 29, 2020*

**Summary**

Supraglacial lake drainage has crucial impacts on surface-to-bed meltwater connection on the Greenland Ice Sheet but remains challenging to quantify. This study uses remote-sensing velocity datasets to constrain the relationship between strain rates and supraglacial lake drainage and to test the hypothesis that transient strain rates drive fast lake drainage. The results show significantly more-extensional background strain rates at moulins associated with fast-draining lakes than at slow-draining or non-draining lake moulins. This study aims to solve an important science question for the Greenland re-search community. I recommend it for publication with some minor changes.

**General Comments**

(1) The structure of the paper is basically clear but can be improved. Some suggestions:

Section 3.4, this section is short so it may not be necessary to divide it into two sub-sections.
*We agree that the brevity makes it not strictly necessary, but we think that sub-dividing this section is helpful.  It also parallels the sub-division of Section 3.5, which is much longer and contains identical sub-sections on "effect of elevation" and "effect of wintertime strain rate".*

Section 3.6, some descriptions belong to methods and should be removed.
*We have moved the descriptions text to Methods (new Section 2.2.2, "Calculation of principal strain rates from GPS station pairs") and expanded the uncertainty formulas (Equations 6–9) in the Methods to improve their applicability to GPS data (Section 2.2.1, "Calculation of principal strain rates").*

Section 3.7, the first paragraph of this section belongs to methods and should be removed.
*We have moved this text to Methods (new Section 2.2.3, "Stacking of strain rates across multiple melt seasons").*

The discussion section presents very insightful ideas but the discussion should be based on the results of this study. I think sections "4.2 Prediction of future lake-drainage events" and "4.3 Parameterizing moulins in ice-sheet models" should be more closely related to the main findings of this study. In other words, these two sections should highlight how the findings of this study can help us better answer the two crucial science questions (lake drainage and new generation of ice

sheet models) rather than broadly introducing these two science questions. This can be done by slightly modifying some words and expressions.

*We have added sentences to Sections 4.2.1 and 4.2.2 to more explicitly tie our extensions to the analysis we performed. We have also added points of discussion to Sections 4.2.3 at the suggestion of the other reviewer. Finally, we inserted more sentences within Section 4.3 to reaffirm how the negative results of our experiments require extensions like the ones we discuss.*

(2) Significance test is widely used in the study. It may be useful to briefly explain how the test was conducted at different parts of the results.

*We have added a new subsection to Methods (Section 2.5, "Statistical tests") that describes the two statistical tests we used.*

(3) The paper concludes that "observational progress in understanding lake drainage initiation will rely on field-based tools such as GPS networks and photogrammetry". I think this should be further discussed. A growing availability of high-resolution satellite imagery (e.g. CubeSat and Landsat-9) provide more frequent observations of supraglacial lakes in future and may mitigate the time gap problem.

*We address the Planet.com CubeSat products in Section 4.2.2 (lines ~679 in the differenced document), which will require improved georeferencing if they are to be useful for measuring velocity fields. We added an assessment of the forthcoming Landsat 9, which will operate in tandem with Landsat 8 much as the Sentinel-1 constellation does, improving the temporal coverage from a 16-day repeat to an 8-day repeat cycle 2 (lines ~676 in the differenced document).*

(4) The study area of this paper is relatively small (~1600 km$^2$) and most cover low elevations (<1400 m). Will the results obtained in this study be applicable for larger areas? particularly when including high-elevation areas. It will be useful to briefly discuss this point.

*Indeed there is likely to be some ice-thickness dependence on lake drainage, perhaps substantial, and perhaps in a way that would not extrapolate well from our study elevation (s <~1500 m). We added a brief discussion of this to Section 4.3 (lines ~793 in the differenced document).*

**Specific Comments**

line 16, Smith et al (2015) found nearly all surface meltwater drain into moulins in the ablation zone of the southwestern GrIS rather than in the western GrIS. I think it is necessary to distinguish these two study areas. It is not clear if all meltwater drains to moulins, particularly at the high-elevation areas of the western GrIS since few river, lake, or moulin maps have been made for this region.

*Here we use "western" as a general term to include the large and widely studied ablation zone on the western flank of the Greenland Ice Sheet. This includes Pâkitsoq (our primary study region) as well as regions north and south of it, including the "southwestern" Greenland Ice Sheet (the Russell Glacier catchment studied by Smith et al. (2015), Doyle et al. (2013), etc.). Both regions have a long history of study, especially Pâkitsoq, whose proximity to Ilulissat allowed extensive study by GEUS in the late 1980s (e.g., the hydrologic map of Thomsen 1988). Indeed it is possible that high-elevation meltwater on the western (or southwestern) Greenland Ice Sheet meets a different fate than lower-elevation meltwater, for*

*instance refreezing in firn, buried lakes, or ice slabs, but we don't know of any reason this should differ between the Pâkitsoq and Russel Glacier regions.*

line 18, on diurnal scale too.

> *We changed "daily" to "diurnal" for clarity (line 18 in the differenced document).*

line 20, add "." before "Our".

> *Fixed (line 20 in the differenced document).*

line 28, "basins of specific supraglacial lakes", do "basins" mean the topographic depressions that host lakes or the upstream contributing catchment area to feed lakes?

> *We mean the topographic depressions. We replaced "basins" with "topographic depressions" to be clearer (line 29 in the differenced document).*

line 42, supraglacial river gauging, streams are narrow and exhibit small contributing areas.

> *We use "supraglacial streams" throughout the manuscript to refer to water-conveying channels on the ice surface. Some authors distinguish between supraglacial "rivers" (larger fluxes) and "streams" (smaller fluxes), but knowledge of relative water fluxes across channels is outside the focus of our remote-sensing-based study.*

line 50, Banwell and Sommers are not appropriate to describe "the next generation of ice sheet models".

> *We agree they do not describe the next generation of models, but these references do suggest the needed improvements in the representation of the ice-sheet hydrologic systems in forthcoming models. We added text to clarify this (line 41 in the differenced document).*

lines 59-61, this sentence is not easy to follow.

> *We removed a few words and added commas to make this simpler (lines ~63 in the differenced document).*

lines 102-103, how to obtain velocity uncertainties?

> *We use the posted uncertainties associated with each velocity product. To clarify, we added the word "posted" (line 112 in the differenced document).*

line 121, panchromatic pixels?

> *We removed the word "pixels" (line 126 in the differenced document).*

line 232, panchromatic band? Multi-spectral bands have lower spatial resolution (2 m).

> *We have updated this section with the correct band-specific resolutions and to use the words "multispectral" and "panchromatic" to be more precise (line ~281 in the differenced document).*

line 258, how is p value calculated?

> *See our response to the earlier "general comment" on significance testing.*

line 266, e3?

> *Fixed (line 320 in the differenced document).*

line 285, are most of these moulins located in topographic depressions that host lakes?
> *Yes; this is addressed in Section 4.1.2 and illustrated, for a single example lake, in Figure 5. We added text to specify this (lines ~354 in the differenced document).*

lines 308-310, how about comparing to Cooley and Christoffersen (2017)?
> *Comparison added (lines ~372 and ~375 in the differenced document).*

lines 320-322, how is p value calculated?
> *See our response to the earlier "general comment" on significance testing.*

line 421, standard deviation 15 days is a very long time because most supraglacial lakes have short lifetime spans. Any implications we can obtain from this long std?
> *Indeed, the drainage date varies significantly across our dataset of 78 lakes and 19 melt seasons. Lakes in our dataset span elevations from roughly 400–1400 meters, which has a significant effect on drainage date ($p<10^{-9}$, Figure 9), and the date of melt onset spans early April through early June from year to year, which also has a significant effect on drainage date ($p<10^{-9}$, Figure 9). The lifetime of a fast-draining lake on the ice sheet is some dozens of days (see example in Figure 2a), so it does seem that a given lake may well be present or absent on a given calendar day from year to year, or that lakes at lower elevations may drain before higher-elevation lakes fill.*

line 428, what does "meaningfully change" mean?
> *By "the values do not meaningfully change" we mean that they fluctuate within the range of their natural variability and within the calculated errors on these data.*

line 591, fix "*Stevens:2015ht".
> *Fixed (line 663 in the differenced document).*

line 723, "meltwater" rather than "melt" delivery to the bed.
> *Fixed (line 824 in the differenced document).*

Figure 4, the moulin elevation colors are not clear for the dots, perhaps change into color ramp? Fix "Bamber:2013 gw" in the figure caption.
> *We've fixed the black-to-gray-to-white tones, which incorrectly stopped at mid-gray, and fixed the typo in the caption. We've also corrected an error in the data for this figure – the Discussion version of the figure incorrectly showed lake drainage dates from an outdated version of the database.*

Figure 8, see the comment for Figure 4.
> *These black-to-gray-to-white tones display correctly on this figure. We experimented with the suggestion to change these grays to a color ramp, but all color ramps we tried obfuscated the red and blue strain rates or made the figure even busier than it already is. Ultimately, we doubled the marker size to make the elevation data more visible.*